# Provable Advantage of Curriculum Learning on Parity Targets with Mixed Inputs

**Emmanuel Abbe**
EPFL
emmanuel.abbe@epfl.ch

**Elisabetta Cornacchia**
EPFL, MIT
ecornacc@mit.edu

**Aryo Lotfi**
EPFL
aryo.lotfi@epfl.ch

## Abstract

Experimental results have shown that curriculum learning, i.e., presenting simpler examples before more complex ones, can improve the efficiency of learning. Some recent theoretical results also showed that changing the sampling distribution can help neural networks learn parities, with formal results only for large learning rates and one-step arguments. Here we show a separation result in the number of training steps with standard (bounded) learning rates on a common sample distribution: if the data distribution is a mixture of sparse and dense inputs, there exists a regime in which a 2-layer ReLU neural network trained by a curriculum noisy-GD (or SGD) algorithm that uses sparse examples first, can learn parities of sufficiently large degree, while any fully connected neural network of possibly larger width or depth trained by noisy-GD on the unordered samples cannot learn without additional steps. We also provide experimental results supporting the qualitative separation beyond the specific regime of the theoretical results.

## 1 Introduction

Starting with easy concepts is an effective way to facilitate learning for humans. When presented with simpler ideas or tasks, individuals can develop a foundation of knowledge and skills upon which more complex concepts can be built. This approach allows learners to gradually gain competence and accelerate learning [EA84, RK90, AKB$^+$97, SGG14].

For machine learning algorithms, this is typically referred to as *Curriculum Learning (CL)* [BLCW09]. Several empirical studies have shown that presenting samples in a meaningful order can improve both the speed of training and the performance achieved at convergence, compared to the standard approach of presenting samples in random order [WCZ21, SIRS22, GBM$^+$17, HW19, PSL15].

The theoretical study of CL is mostly in its infancy. While some works have analytically shown benefits of CL for certain targets [SMS22, WCA18, MKAS21, CM23] (see Section 1.1 for a discussion on these), it remained open to provide a separation result between curriculum training and standard training on a common sampling distribution. Specifically, one seeks to demonstrate how a standard neural network, trained using a gradient descent algorithm on a given dataset and horizon, can successfully learn the target when certain samples are presented first during training, whereas learning does not occur if samples are presented in a random order.

This paper provides such an instance. We focus on the case of parities, as a classical case of challenging target for differentiable learning. In the experimental part of the paper, we report results suggesting that a similar picture can hold for other large leap functions as defined in [ABAM23], although more investigations are required to understand when and how curriculum learning should be developed for more general functions.

We assume that the network is presented with samples $(x, \chi_S(x))$, with $x \in \{\pm 1\}^d$ and $\chi_S(x) := \prod_{i \in S} x_i$, for some set $S \subseteq \{1, \ldots, d\}$ of fixed size. We consider datasets where a small fraction $\rho$ of

the inputs are *sparse*, meaning they have, on average, few negative coordinates and mostly positive coordinates, and the remaining inputs are *dense*, with an average of half negative and half positive coordinates. We refer to Section 2 for the definition of the input distribution. Parities seem to be promising candidates for theoretical investigations into CL, as they are hard to learn in the statistical query learning model [Kea98], or on dense data by any neural network trained with gradient-based methods having limited gradient precision [AS20], but they are efficiently learnable on sparse data by regular architectures [MKAS21] (see Section 1.1 for further discussion).

We say that a neural network is trained *with curriculum* if it goes through an initial phase where it is exposed to batches consisting only of sparse samples (see discussion below on the interpretation of these as 'simpler' examples). Following this phase, the network is trained using batches sampled from the entire dataset. We compare this curriculum training approach with standard training, where the network receives batches of samples selected randomly from the entire dataset at each step. Our intuition is as follows. If the fraction of sparse samples is small, without curriculum, the network is exposed to batches of mostly uniform inputs, and the contribution of sparse inputs in estimating the gradients will be limited. As a result, learning parities on the mixed distribution without curriculum is roughly as hard as learning parities on dense inputs (Theorem 2). On the contrary, if the network is initially exposed to batches consisting of sparse inputs, it can efficiently identify the support of the parity, and subsequent fitting using batches sampled from the entire dataset enables learning the target with fewer training steps compared to standard training (Theorem 1).

In particular, we show that if the fraction of sparse samples $\rho$ is small enough, (e.g., $\rho < d^{-4.5}$, where $d$ is the input dimension), a 2-layer fully connected network of size $\theta(d)$ trained by layerwise SGD (or noisy-SGD, see Def. 1) with curriculum can learn any target $k$-parity, with $k$ bounded, in $T = \theta(d)$ steps, while the same network (or any network of similar size) trained by noisy-SGD without curriculum cannot learn the target in less than $\Omega(d^2)$ steps. We refer to Section 2 for a more detailed explanation of our results. While our theoretical results use simplifying assumptions (e.g., layerwise training), our experiments show separations between curriculum and standard training in a broader range of settings.

## 1.1 Related Literature

Curriculum Learning (CL) was first introduced in the machine learning context by the seminal work of [BLCW09]. This paper along with many other subsequent works presented empirical evidence that training neural networks with a curriculum can be beneficial in both the training speed and the performance achieved at convergence in certain instances in different domains such as computer vision [JMMH14, GHZ+18, CG15, STD19], natural language processing (NLP) [KB17, GBM+17, ZKK+18, PSN+19], and reinforcement learning [FHW+17, NPL+20]. We refer to relevant surveys [WCZ21, SIRS22] for a more comprehensive list of applications of CL in different domains. In general, defining a complexity measure on samples and consequently an ordering of training inputs is not trivial. Some works determine the easiness of samples based on predefined rules (e.g., shape as used in [BLCW09]), while others also use the model's state and performance to evaluate the samples dynamically as in self-paced learning (SPL) [KPK10]. In this work, we follow the former group as we define the hardness of samples based on the number of $-1$ bits, i.e., the Hamming weight. Note that in parity targets, $+1$ is the identity element. Thus, one can view the number of $-1$ bits as the *length* of the sample. In this sense, our work resembles [SAJ10, ZS14, KB17, PSN+19] which use the length of the text inputs as a part of their curriculum.

Despite the considerable empirical work on CL, there appears to be a scarcity of theoretical analysis. Few theoretical works focused on parity targets, as in this paper. While parities are efficiently learnable by specialized algorithms that have access to at least $d$ samples (e.g. Gaussian elimination over the field of two elements), they are hard to learn under the uniform distribution (i.e. on dense data) by any neural network trained with gradient-based methods with limited gradient precision [AS20, AKM+21]. Parities on sparse data can be learned more efficiently on an active query model, which can be emulated by a neural network [AS20]. However, such emulation networks are far from the practical and homogeneous networks, and distribution class-dependent. A natural question that has been considered is whether more standard architectures can also benefit from a simpler data distribution. In [MKAS21], the authors make a first step by showing that parities on sparse inputs are efficiently learnable on a 1-layer network with a parity module appended to it, with a one-step argument. In [DM20], it is shown that for a two-layer fully connected network, sparse inputs help but

with a variant of the distribution that allows for leaky labels. [ABLR23] obtains experimental results for a curriculum algorithm presenting sparse samples first with heuristics for regular networks and hyperparameters, but without proofs. In [CM23], the authors prove that a two-layer fully connected net can benefit from sparse inputs, with a one-step gradient argument requiring a large learning rate. This paper improves on [CM23] not only in terms of having more natural training settings and hyperparameters (i.e., we use SGD with bounded batch size and bounded learning rate), it is also establishing a separation between learning with and without curriculum on a common sampling distribution, while [CM23] does not (in fact in their work curriculum involves both sparse and dense inputs while standard training involves dense inputs only). To the best of our knowledge, our paper is the first work that establishes a rigorous separation between training on adequately ordered samples and randomly ordered samples drawn from the same distribution, and our theorem gives a condition on the mixture under which this separation holds (interestingly the separation does not take place for all mixtures parameters). Beyond parities, [SMS22] studies a teacher-student model, where the target depends on a sparse set of features and where the variance on the irrelevant features. Furthermore, [WCA18, WA20] show that on some convex models, CL provides an improvement in the speed of convergence of SGD. In contrast, our work covers an intrinsically non-convex problem.

## 2    Setting and Informal Contributions

We consider the problem of learning a target function $f : \{\pm 1\}^d \to \{\pm 1\}$ that belongs to the class of $k$-parities on $d$ bits, i.e.,

$$f \in \{\chi_S(x) = \prod_{j \in S} x_j : S \subseteq [d], |S| = k\}, \tag{1}$$

where we denoted by $[d] := \{1, \ldots, d\}$. We assume that the network has access to a dataset $(X, Y) = \{(x^s, y^s)\}_{s \in [m]}$ where the inputs $x^s$ are sampled i.i.d. from a *mixed* distribution defined as follows:

$$\mathcal{D}_{\text{mix}} = \rho \mathcal{D}_\mu + (1 - \rho) \mathcal{D}_u, \tag{2}$$

where $\mathcal{D}_\mu = \text{Rad}(\frac{\mu+1}{2})^{\otimes d}$[1], for some $\mu \in [-1, 1]$, $\mathcal{D}_u = \text{Unif}\{\pm 1\}^d$[2] and $\rho \in [0, 1]$ is some fixed parameter, and $y^s = f(x^s)$. We will assume $\mu \in (0, 1)$. In other words, we assume that the dataset contains approximately $\rho m$ samples with in average $\frac{1-\mu}{2} d$ negative bits (sparse samples), and $(1 - \rho)m$ samples with in average half negative bits (dense samples). We consider training the neural network with stochastic gradient descent, with or without gradient noise (SGD or noisy-SGD, see Def. 1).

**Informal Description of the Curriculum and Standard Training Algorithms.** We compare two training strategies: standard training and curriculum training. In the standard training, at each step, mini-batches are drawn from the whole dataset. For curriculum training, we first need to isolate the set of sparse inputs in our dataset. Let us denote this set by $X_1 := \{x^s \in X : H(x^s) < (\frac{1}{2} - \frac{\mu}{4})d\}$, where $H(x) := \sum_{j \in [d]} \mathbb{1}(x_j = -1)$ denotes the *Hamming weight* of $x$, which is the count of negative bits in $x$, and by $Y_1 = \{y^s \in Y : x^s \in X_1\}$ the corresponding set of labels. Our curriculum strategy consists of training on only sparse samples for the first $T_1$ steps, where $T_1 \leq T$ is some well-chosen time horizon and $T$ is the total number of training steps. After this initial phase, the network is trained on samples belonging to the whole dataset. To sum up, we compare the following two training strategies:

- (standard training): At all $t \in [T]$, mini-batches are sampled from $(X, Y)$;
- (curriculum training): At $t \leq T_1$, mini-batches are sampled from $(X_1, Y_1)$ and at $T_1 < t \leq T$ mini-batches are sampled from $(X, Y)$.

**Informal Description of the Positive and Negative Results.** In this paper, we prove that if the fraction of sparse samples $\rho$ is small enough, there is a separation in the number of training steps needed to learn the target parity up to a given accuracy. We say that an algorithm on a neural network

---

[1]Rad denotes the Rademacher distribution, i.e., we say that $x \sim \text{Rad}(p)$, for $p \in [0, 1]$, if $\mathbb{P}(x = 1) = 1 - \mathbb{P}(x = -1) = p$.

[2]Unif denotes the uniform distribution. In particular, $\mathcal{D}_u = \mathcal{D}_0$.

learns a target function $f\colon \{\pm 1\}^d \to \{\pm 1\}$ up to accuracy $1 - \epsilon$, for $\epsilon \ge 0$, in $T$ steps, if it outputs a network $\mathrm{NN}(x; \theta^T)$ such that:

$$\mathbb{P}_{x \sim \mathcal{D}_{\mathrm{mix}}}(\mathrm{sgn}(\mathrm{NN}(x; \theta^T)) = f(x)) \ge 1 - \epsilon, \tag{3}$$

where $\mathrm{sgn}(.)$ denotes the sign function. In our main positive result, we consider training with the *covariance loss* (Def. 2) and we refer to Remark 2 for considerations on other losses. Let us state our main positive result informally.

**Theorem 1** (Theorem 3, Informal)**.** *Consider the input distribution $\mathcal{D}_{\mathrm{mix}}$ with mixture parameters $\rho = \tilde{O}(d^{-1/2})^3$, $\mu = \theta(1)$. Let $\epsilon > 0$. Then, a 2-layer* ReLU *neural network of size $O(d)$ initialized with an isotropic distribution, trained with the covariance loss, standard (i.e., non-diverging) learning rate schedule and either (i) a layer-wise curriculum-noisy-SGD algorithm with gradient range $A = \tilde{O}(\sqrt{d})$, noise-level $\tau = \tilde{O}(d^{-1})$ and sparse samples first, or (ii) a layer-wise curriculum-SGD algorithm with sparse samples first, can learn any $k$-parities, $k = \theta(1)$, up to accuracy at least $1 - \epsilon$ in $T = \tilde{O}(d)/\epsilon^2$ steps.*

We further show an additional positive result valid for a layer-wise curriculum SGD with the hinge loss, which holds for large batch size and diverging learning rate (Theorem 4). On the flip side, we have the following.

**Theorem 2** (Theorem 5, Informal)**.** *Consider the input distribution $\mathcal{D}_{\mathrm{mix}}$ with mixture parameters $\rho = \tilde{O}(d^{-3.5-\delta})$, $\delta > 0$, and $\mu = \theta(1)$. Then, any neural network of size $\tilde{O}(d)$ (any depth), with any initialization and activation, trained by a noisy-SGD algorithm (layer-wise or joint) with gradient range $A = \tilde{O}(\sqrt{d})$, noise-level $\tau = \tilde{\Omega}(d^{-1})$, batch size $B = \tilde{\Omega}(\rho^{-2})$, any almost surely differentiable loss function and any learning rate schedule, will need at least $T = \tilde{\Omega}(\epsilon d^{1+\delta})$ steps to achieve accuracy $\frac{1}{2} + \epsilon$ for any $k$-parities ($k \ge 6$).*

**Conclusion of Results.** When using noisy-SGD, the curriculum learning algorithm that takes sparse samples first, for the first layer training, and then full samples, can learn in a regime of hyperparameters and training steps (Theorem 1) that is contained in the hyperparameter regime of the negative result (Theorem 2), for which it is shown that the absence of curriculum cannot learn without additional training steps. Further, it is shown in part (ii) of the first theorem, that using SGD rather than noisy-SGD still allows to learn with the curriculum training; this result does however not come with a counter-part for the lower-bound, i.e., we cannot show that SGD without curriculum could not learn such parities, as there is currently no proof technique to obtain rigorous lower-bounds for SGD algorithms (as for the case of noiseless honest-SQ algorithms); see for instance discussions about this in [AKM+21, ABA22]. However, it is not expected that SGD can do better than noisy-SGD on 'regular' neural networks [ABA22, ABAM22, ABAM23].

## 3 Positive Results for Curriculum Training

**The Algorithm.** Let us define the noisy-SGD algorithm, that is used in both our positive and negative results.

**Definition 1** (Noisy-SGD)**.** *Consider a neural network $\mathrm{NN}(.; \theta)$, with initialization of the weights $\theta^0$, and a dataset $(X, Y) = \{(x^s, y^s)\}_{s \in [m]}$. Given an almost surely differentiable loss function $L$, the updates of the noisy-SGD algorithm with learning rate $\gamma_t$ and gradient range $A$ are defined by*

$$\theta^{t+1} = \theta^t - \gamma_t \left( \frac{1}{B} \sum_{s=1}^{B} \left[ \nabla_{\theta^t} L(\mathrm{NN}(x^{s,t}; \theta^t), y^{s,t}, x^{s,t}) \right]_A + Z^t \right), \tag{4}$$

*where for all $t \in \{0, \dots, T-1\}$, $Z^t$ are i.i.d. $\mathrm{Unif}[-\tau, \tau]$, for some noise level $\tau$, and they are independent from other variables, $B$ is the batch size, and $[z]_A := \mathrm{argmin}_{|y| \le A} |y - z|$, i.e., whenever the gradient exceeds $A$ (resp. $-A$) it is rounded to $A$ (resp. $-A$).*

For brevity, we will write $L(\theta^t, y, x) := L(\mathrm{NN}(x; \theta^t), y, x)$. We assume that for all $s \in [B]$, $(x^{s,t}, y^{s,t})$ are chosen from $(X_t, Y_t)$, where $(X_t, Y_t)$ can be either the whole dataset $(X, Y)$, or a subset of it. We include the noise $Z^t$ and the gradient range $A$ to cover the noisy-SGD algorithms,

---

$^3 \tilde{O}(d^c) = O(d^c \mathrm{poly}(\log(d)))$, for any $c \in \mathbb{R}$.

used in SQ lower bounds e.g. in [AS20, AKM+21, MKAS21, ABA22, ACHM22]. Note that if we set $\tau = 0$ and $A$ large enough, we recover the standard SGD algorithm without noise.

We assume that our dataset is drawn from the mixed distribution defined in (2), with parameters $\rho, \mu$. For the purposes of this section, we assume $\rho = \tilde{O}(d^{-1/2})$ and $\mu \in (0, 1)$. As a preliminary step, we isolate the sparse inputs from our dataset, by mean of Hamming weight. In particular, we define $X_1 := \{x^s \in X : H(x^s) < (\frac{1}{2} - \frac{\mu}{4})d\}$, where $H(x) := \sum_{j \in [d]} \mathbb{1}(x_j = -1)$ denotes the *Hamming weight* of $x$, and $Y_1 = \{y^s \in Y : x^s \in X_1\}$. Note that, due to the concentration of Hamming weight, for $d$ large enough, there will be approximately $m\rho$ samples in $X_1$ (see Lemma 1). We assume that the training set is large enough, such that each sample is observed at most once during training.

In this section, we consider a 2-layer neural network $\mathrm{NN}(x; \theta)$, with $\theta = (a, w, b)$, defined by: $\mathrm{NN}(x; \theta) := \sum_{i=1}^{N} a_i \sigma(w_i x + b_i)$, where $\sigma(x) = \mathrm{ReLU}(x) := \max\{x, 0\}$. We assume that the number of hidden units is $N \geq k + 1$, where $k$ is the degree of the target parity. We initialize $w_{ij}^0 = 0$, $a_i^0 = \kappa$ and $b_i^0 = \Delta d + 1$, for some $\kappa, \Delta > 0$, for all $i, j$. We refer to $\kappa$ and $\Delta$ as the second layer initialization scale and the bias initialization scale, respectively. We adopt a layer-wise curriculum training approach. During the initial phase of training, we exclusively train the first layer of the network using samples taken from $(X_1, Y_1)$. Furthermore, we project the weights of the first layer, to guarantee that they stay bounded. In the second phase, we train only the second layer using the entire merged dataset. We refer to Algorithm 1 in Appendix A for the pseudo-code of the algorithm used in our proof. In the context of standard learning without curriculum, similar layer-wise training strategies are considered in e.g., [MSS20, BEG+22, ABAM23]. [CM23] uses a layer-wise curriculum training, but with a one-step argument that requires a diverging learning rate. In our analysis, we keep the bias weights fixed to specific values during the whole training. We believe that one could extend the result to include scenarios with more general initializations and where the bias weights are trained, but this would require further technical work.

**The Covariance Loss.** We train $\mathrm{NN}(x; \theta)$ using noisy-SGD with a specific loss function, namely the *covariance loss*, that we define here. The covariance loss appears in [CM23] with a slightly different definition: in particular, the following definition does not depend on the average estimator output, as opposed to their definition.

**Definition 2** (Covariance Loss). *Let $(X, Y) = \{(x^s, y^s)\}_{s \in [m]}$ be a dataset, where for all $s$, $x^s \in \mathcal{X}$ and $y^s \in \{\pm 1\}$, and let $\bar{y} = \frac{1}{m} \sum_{s \in [m]} y^s$. Assume $|\bar{y}| < 1$. Let $\hat{f} : \mathcal{X} \to \{\pm 1\}$ be an estimator. We define the covariance loss as*

$$L_{\mathrm{cov}}(\hat{f}, y^s, x^s) := \left(1 - y^s \bar{y} - \hat{f}(x^s)(y^s - \bar{y})\right)_+, \tag{5}$$

*where $(a)_+ := \max\{a, 0\}$.*

The covariance loss can be applied to classification datasets with non-negligible fractions of $+1$ and $-1$ labels. In Proposition 1, we show that a small covariance loss implies a small classification error.

**Proposition 1.** *Let $(X, Y) = \{(x^s, y^s)\}_{s \in [m]}$ be a dataset, where for all $s$, $x^s \in \mathcal{X}$ and $y^s \in \{\pm 1\}$. Let $\bar{y} = \frac{1}{m} \sum_{s \in [m]} y^s$ and assume $|\bar{y}| < 1 - \delta$, for some $\delta > 0$. Let $\hat{f} := \mathcal{X} \to \{\pm 1\}$ be an estimator. If $\frac{1}{m} \sum_{s \in [m]} L_{\mathrm{cov}}(\hat{f}, x^s, y^s) < \epsilon$, then $\frac{1}{m} \sum_{s \in [m]} \mathbb{1}(\mathrm{sgn}(\hat{f}(x^s)) \neq y^s) < \frac{\epsilon}{\delta}$.*

The proof of Proposition 1 can be found in Appendix D.

*Remark* 1. We remark that for balanced datasets ($\bar{y} = 0$), the covariance loss coincides with the hinge loss. On the other hand, note that the right hand side of (5) can be rewritten as $\left((1 - y^s \bar{y}) \cdot (1 - y^s \hat{f}(x^s))\right)_+$. Thus, in some sense, the samples in the under-represented class (where $y^s \bar{y} \leq 0$) are assigned a higher loss compared to those in the over-represented class (where $y^s \bar{y} \geq 0$).

**Main Theorem.** We are now ready to state our main theorem.

**Theorem 3** (Main Positive Result). *Let $\mathrm{NN}(x; \theta)$ be a 2-layer* ReLU *neural network with $N \geq k + 1$ hidden units, 2nd layer init. scale $\kappa > 0$ and bias init. scale $\Delta > 0$. Let $\mathcal{D}_{\mathrm{mix}}$ be a mixed distribution, as defined in (2), with parameters $0 < \rho \leq \frac{\Delta}{4N} \frac{\sqrt{\log(d)}}{\sqrt{d}}$, and $\mu \in (0, 1)$. Assume we have access to*

*a dataset* $(X, Y) = \{x^s, y^s\}_{s \in [m]}$, *where* $x^s \overset{iid}{\sim} \mathcal{D}_{\mathrm{mix}}$, $y^s = \chi_S(x^s)$, *with* $|S| = k$. *Assume* NN *is trained according to the layer-wise curriculum-SGD algorithm (Algorithm 1), with the covariance loss, batch size* $B$, *noise level* $\tau$ *and gradient range* $A$.

*For all* $\epsilon > 0$, *there exist* $C, C_1, C^* > 0$ *such that if* $\kappa \leq 1/N(2\Delta d + 2)$, $\tau \leq \kappa$, $A = \tilde{\Omega}(\sqrt{d})$, $m \geq 4C^* \log(3d) d\rho^{-1} B$, *and if*

$$T_1 = \frac{2C\Delta^2 d \log(d)^2}{L_\mu^2}, \qquad\qquad \gamma_1 = \frac{L_\mu}{C\Delta\kappa d \log(d)^2}, \qquad (6)$$

$$T_2 = \frac{256(C_1 + 2\Delta k + \tau)^2 N^2 d}{\epsilon^2 \Delta^2 \log(d)}, \qquad \gamma_2 = \frac{\epsilon \log(d)}{8(C_1 + 2\Delta k + \tau)^2 dN}, \qquad (7)$$

*where* $L_\mu = \mu^{k-1} - \mu^{k+1}$, *then, with probability at least* $1 - 6Nd^{-C^*}$, *the algorithm outputs a network such that* $\frac{1}{m} \sum_{s \in [m]} L_{\mathrm{cov}}(\theta^{T_1 + T_2}, x^s, y^s) \leq \epsilon$.

The following corollary is a direct consequence of Theorem 3 and Proposition 1.

**Corollary 1.** *Let the assumptions of Theorem 3 be satisfied and assume that* $\tau = O(\kappa)$ *and* $\Delta, k$ *constants. There exists a 2-layer* ReLU *network of size at most* $(d + 2)(k + 1)$ *and initialization invariant to permutations of the input neurons, such that for all* $\epsilon > 0$ *the layer-wise curriculum noisy-SGD algorithm with gradient range* $A = \tilde{O}(\sqrt{d})$, *after at most* $T = \tilde{O}(d)$ *steps of training with bounded learning rates, outputs a network such that* $\mathbb{P}_{x \sim \mathcal{D}_{\mathrm{mix}}}(\mathrm{sgn}(\mathrm{NN}(x; \theta^T)) = f(x)) \geq 1 - \epsilon$, *with high probability.*

*Remark* 2. The choice of the covariance loss is motivated by a simplification of the proof of Theorem 3. Indeed, with this loss, we can show that in the first part of training the weights incident to the relevant coordinates move non-negligibly, while the other weights stay close to zero (see proof outline). If we used another loss, say the hinge loss, all weights would move non-negligibly. However, we believe that one could still adapt the argument to cover the hinge loss case, by adding another projection to the iterates. In the following, we also give a positive result for the hinge loss, with a different proof technique, that holds for large batch size and large learning rate (Theorem 4).

*Remark* 3. While we show that with appropriate hyperparameters, the layer-wise curriculum-SGD can learn $k$-parities in $\tilde{O}(d)$ steps, we need to notice that to implement the curriculum, one has to first retrieve $(X_1, Y_1)$. This involves computing the Hamming weight of $d/\rho$ samples, which adds computational cost if the curriculum learning includes this part. Nevertheless, in the experiments that we performed, the recovery of $(X_1, Y_1)$ was consistently efficient, with an average of less than 1 second running time for a dataset of 10 million samples of dimension 100. On the other hand, the reduction of training steps due to the curriculum yields significant time savings.

*Remark* 4. Theorem 3 provides an upper bound on the sample complexity required to learn using a curriculum, of $\tilde{O}(dB/\rho)$ for SGD with batch size $B$. Consequently, while the number of iterations needed for learning remains unaffected by $\rho$, the sample complexity increases as $\rho$ decreases. Additionally, we do not currently have a lower bound on the sample complexity for learning parities with standard training with offline SGD. However, our experiments in Section 5 demonstrate a noticeable difference in the sample complexity required for learning with and without a curriculum. These findings may encourage future work in establishing a theoretical advantage in the number of samples needed to learn with curriculum.

*Remark* 5. We further remark that the proof method of Theorem 3 yields a tighter upper bound on curriculum learning's sample complexity than the 1-step argument used in [CM23][Thm 4]. Specifically, using the technique of [CM23][Thm 4], we achieve an upper bound of $\tilde{O}(d^2/\rho)$, as opposed to the $\tilde{O}(d/\rho)$ given by Theorem 3 for SGD with bounded batch size.

*Remark* 6. $T_1$ depends on the degree of the parity $k$ through the parameter $L_\mu$. We believe that such dependence cannot be removed, however, we remark that choosing $\mu$ sufficiently close to 1 (e.g. $\mu = 1 - \frac{1}{k}$) allows bounding $L_\mu$ independently from $k$. Indeed, our experiments (Figure 4) show mild dependence on the parity degree for curriculum learning with large $\mu$.

**Proof Outline.** The proof of Theorem 3 follows a similar strategy as in [AGJ21, ABAM23, TV19]. We decompose the dynamics into a drift and a martingale contribution, and we show that the drift allows us to recover the support of the parity in $\tilde{O}(d)$ steps, while in the same time horizon, the martingale contribution remains small, namely of order $O(1/\sqrt{d \log(d)})$. In particular, we show that

after the first phase of training, with high probability, all weights corresponding to coordinates in the parity's support are close to $\Delta$ and the others are close to zero. The fit of the second layer on the whole dataset corresponds to the analysis of a linear model and follows from standard results on the convergence of SGD on convex losses (e.g., in [SSBD14]). The formal proof can be found in Appendix A.

**Hinge Loss.** While the use of the covariance loss simplifies considerably the proof of Theorem 3, we believe that the advantages of curriculum extend beyond this specific loss. Here, we present a positive result that applies to a more commonly used loss function, namely the hinge loss.

**Theorem 4** (Hinge Loss). *Let $\mathrm{NN}(x; \theta) = \sum_{i=1}^{N} a_i \sigma(w_i x + b_i)$ be a 2-layer fully connected network with activation $\sigma(y) := \mathrm{Ramp}(y)$ and $N = \tilde{\theta}(d^2 \log(1/\delta))$. Assume we have access to a dataset $(X, Y) = \{(x^s, y^s)\}_{s \in [m]}$, with $x^s \overset{iid}{\sim} \mathcal{D}_{\mathrm{mix}}$, with parameters $\rho > 0, \mu \in (0, 1)$, and $y^s = \chi_S(x)$, for some target parity $\chi_S$, with $|S| = k$. Then, there exists an initialization and a learning rate schedule, such that for any target parity $\chi_S$ and for any $\epsilon > 0$, the layer-wise curriculum SGD on the hinge loss with batch size $B = \tilde{\theta}(d^{10}/\epsilon^2 \log(1/\delta))$, noise level $\tau = \tilde{\theta}(\frac{\epsilon \mu^k \log 1/\delta}{d^6})$, $T_1 = 1$ and $T_2 = \tilde{\theta}(d^6/\epsilon^2)$ with probability $1 - 3\delta$, outputs a network such that $\mathbb{P}_{x \sim \mathcal{D}_{\mathrm{mix}}}(\mathrm{sgn}(\mathrm{NN}(x; \theta^T)) = f(x)) \geq 1 - \epsilon$.*

As opposed to Theorem 3, Theorem 4 only holds for SGD with large batch size and unbounded learning rate schedules. The proof of Theorem 4 follows a 1-step argument, similarly to [CM23], and it is deferred to Appendix B.

## 4 Negative Result for Standard Training

In this section, we present a lower bound for learning $k$-parities on the mixed distribution of eq. (2), without curriculum.

**Theorem 5** (Main Negative Result). *Assume we have access to a dataset $(X, Y) = \{(x^s, y^s)\}_{s \in [m]}$, with $x^s \overset{iid}{\sim} \mathcal{D}_{\mathrm{mix}}$, with parameters $\rho = o_d(1), \mu \in (0, 1)$, and $y^s = \chi_S(x)$, with $|S| = k$. Let $\mathrm{NN}(x; \theta)$ be a fully-connected neural network of size $P$, with initialization that is invariant to permutations of the input neurons. Then, the noisy-SGD algorithm with noise level $\tau$, gradient range $A$, batch size $B$, any learning rate schedule, and any loss function, after $T$ steps of training without curriculum, outputs a network such that*

$$\mathbb{P}_{x \sim \mathcal{D}_{\mathrm{mix}}}(\mathrm{sgn}(\mathrm{NN}(x; \theta^T)) = f(x)) \leq \frac{1}{2} + \frac{TPA}{\tau} \cdot \left( \binom{d}{k}^{-1} + C_k \rho^2 \mu^{4k} + \frac{1}{B} \right)^{1/2}, \quad (8)$$

*where the probability is over $x \sim \mathcal{D}_{\mathrm{mix}}$ and any randomness in the algorithm, and where $C_k$ is a constant that depends on $k$.*

*Remark 7.* For the purposes of Theorem 5, the network can have any fully connected architecture and any activation such that the gradients are well-defined almost everywhere. Furthermore, the loss can be any function that is differentiable almost everywhere.

Theorem 5 follows from an SQ-like lower bound argument and builds on previous results in [AS20][Theorem 3]. We defer the formal proof to Appendix C. Theorem 5 implies the following corollary.

**Corollary 2.** *Under the assumptions of Theorem 5, if $\rho = \tilde{O}(d^{-3.5-\delta})$ and $k \geq 6$, any fully connected network of size at most $\tilde{O}(d)$ with any permutation-invariant initialization and activation, trained by the noisy-SGD algorithm with batch size $B = \tilde{\Omega}(\rho^{-2})$, noise-level $\tau = \tilde{O}(1/d)$ and gradient range $A = \tilde{O}(\sqrt{d})$, after $T = \tilde{O}(d)$ steps of training, will output a network such that $\mathbb{P}_{x \sim \mathcal{D}_{\mathrm{mix}}}(\mathrm{sgn}(\mathrm{NN}(x; \theta^T)) = f(x)) \leq \frac{1}{2} + Cd^{-\delta} \log(d)^c$, for some $c, C > 0$.*

In particular, in Corollary 2, the network can have any width and depth (even beyond 2 layers), as long as the total number of parameters in the network scales as $\tilde{O}(d)$ in the input dimension. Combining Corollary 1 and Corollary 2, we get that if $\rho = \tilde{O}(d^{-3.5-\delta})$, for some $\delta > 0$, and $k \geq 6$, and for $\epsilon = 1/10$, say, a 2-layer fully connected network can learn any $k$ parities up to accuracy $1 - \epsilon$ in $T = \theta(d)$ steps with curriculum, while the same network in the same number of steps without curriculum, can achieve accuracy at most $1 - 2\epsilon$, for $d$ large enough.

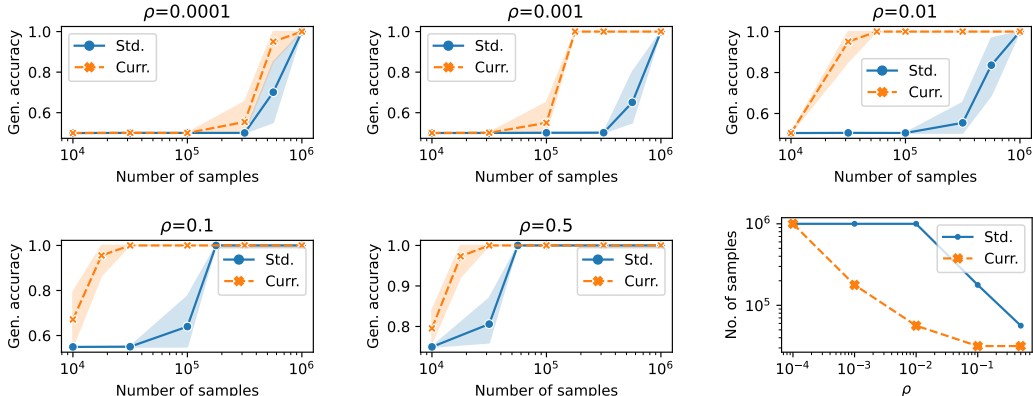

Figure 1: Comparison of the performance of the curriculum strategy and standard training for different training set sizes and values of $\rho$. In the bottom-left plot, we report the number of samples needed to achieve accuracy close to $1$ for different values of $\rho$ based on the other plots.

*Remark* 8. While our results show a first separation in the number of training steps between curriculum and standard training, we do not claim that the given range of $\rho$ is maximal. For instance, one could get a tighter negative bound by considering Gaussian, instead of uniform, noise in the noisy-SGD iterates. However, getting a positive result with the curriculum for Gaussian noise would require a more involved argument. Our experiments show separation in the number of training steps and in the sample complexity for different values of $\rho$.

## 5 Experiments

In this section, we present empirical results demonstrating the benefits of the proposed curriculum method.[4] We use a multi-layer perceptron (MLP) with 4 hidden layers of sizes $512$, $1024$, $512$, and $64$ as our model. We optimize the model under the $\ell_2$ loss using mini-batch SGD with batch size $64$. We also train all layers jointly. Each of our experiments is repeated with 10 different random seeds, and the results are reported with $95\%$ confidence intervals. Note that generalization error is always computed based on $\mathcal{D}_{\mathrm{mix}}$. Here, we mostly focus on the common settings not covered by our theoretical results. We present additional results on other architectures, namely, two-layer mean-field networks [MMN18] and Transformers [VSP+17] and covariance and hinge loss functions in Appendix E.2.

**Number of Samples.** First, we focus on the difference in sample complexity, between curriculum and standard training. We consider learning $f(x_1, \ldots, x_{100}) = x_1 x_2 x_3 x_4 x_5$, with inputs sampled from a mixed distribution with $\mu = 0.98$ and different values of $\rho = 10^{-4}, 10^{-3}, 10^{-2}, 0.1, 0.5$. For each $\rho$, we train the network on training sets of different sizes (between $10^4$ and $10^6$) with and without curriculum. When using the curriculum, we first select the sparse samples as those with Hamming weight smaller than $(\frac{1}{2} - \frac{\mu}{4})d$, and we train on those until convergence (specifically, training loss smaller than $10^{-2}$). We then train the model on the entire training set until convergence. Similarly, when the curriculum is not used, the model is trained using all the available samples until convergence.

Figure 1 shows the validation accuracy achieved for different values of $\rho, m$. It can be seen that for some values of $\rho$, there is a reduction in the number of samples needed to achieve validation accuracy close to $1$ when the curriculum is used. For the purpose of visualization, in Figure 1 (bottom-right), we sketch the number of samples needed to achieve accuracy close to $1$ for different values of $\rho$, based on the previous plots. We notice that the sample complexity decreases as $\rho$ is increased, for both curriculum and standard training. Among the $\rho$ values that we tried, the gap between the two training strategies is maximal for $\rho = 0.01$, and it decreases as $\rho$ gets larger or smaller. Also, note

---

[4]Code: `https://github.com/aryol/parity-curriculum`

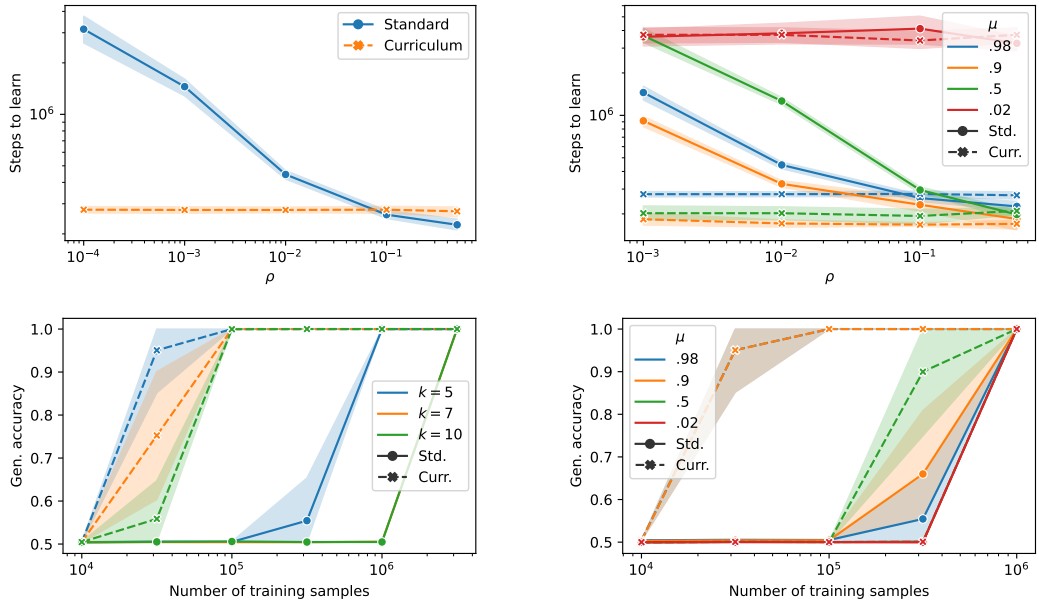

Figure 2: (Top-left) Number of training steps needed for learning 5-parity for different values of $\rho$. The gap between curriculum and standard training decreases as $\rho$ increases. (Top-right) The number of training steps for different values of $\mu$ based on $\rho$. (Bottom-right) Performance for different values of $\mu$ and different sample sizes. Curriculum boosts the learning for large values of $\mu$. (Bottom-left) Performance for different parity functions. (The curves of standard training for $k = 7$ and $k = 10$ overlap.) The benefit of curriculum potentially increases with the degree of the parity.

that for small values of $\rho$ the sample complexity for standard training is constant and equals $10^6$. We note that with $10^6$ samples, we can learn $f(x)$ even with uniform inputs ($\rho = 0$).

**Number of Training Steps.** Now we focus on the number of iterations needed for learning the same $f$ as above. In order to study the number of iterations independently of the number of samples, we draw fresh mini-batches at each step. When using the curriculum, we sample from $\mathcal{D}_\mu$ in the initial phase and from $\mathcal{D}_{\text{mix}}$ in the second phase, while when using standard training we sample from the mixed distribution at each step. In all cases, we train until convergence. The number of iterations needed for learning $f$ for different values of $\rho$ is depicted in Figure 2 (top-left). It can be seen that the number of training steps needed for learning with the curriculum is constant w.r.t. $\rho$. In contrast, for standard training, the number of iterations increases with the decrease of $\rho$. Both these observations are consistent with our theoretical results. In particular, for small values of $\rho$, the curriculum significantly reduces the number of training steps, while for large $\rho$ it provides little or no benefit, and for $\rho = 0.5$, it is harmful.

**Dependency on $\mu$ and $k$.** Next, we investigate the dependency of our results on $\mu$ and on the degree of parity $k$. For evaluating dependency on $\mu$, we consider the function $f$ as above and different values of $\mu$. First, we consider the number of training steps needed for learning $f$ based on $\rho$. The results are shown in Figure 2 (top-right). It can be seen that while the curriculum is not beneficial for $\mu = 0.02$ (where $\mathcal{D}_\mu$ is close to $\mathcal{D}_u$ and all samples are dense), it can significantly reduce the number of training steps for moderate and large values of $\mu$. Further, we fix $\rho = 0.01$ and compare the performance of the curriculum and standard training for different training set sizes. The results are demonstrated in Figure 2 (bottom-right). It is shown that for large values of $\mu$ such as $0.98$ and $0.90$, there is a gap in the number of samples needed for achieving generalization accuracy close to $1$ between curriculum and standard training. On the other hand, for smaller values of $\mu$, this gain is diminished. Nonetheless, note that the relationship between the performance of the curriculum method and $\mu$ is not monotonous (e.g., see the number of steps plot), and this is indeed reflected in our analysis. More precisely, in Theorem 1, $T_1$ scales with $1/L_\mu^2$ where $L_\mu = \mu^{k-1} - \mu^{k+1}$.

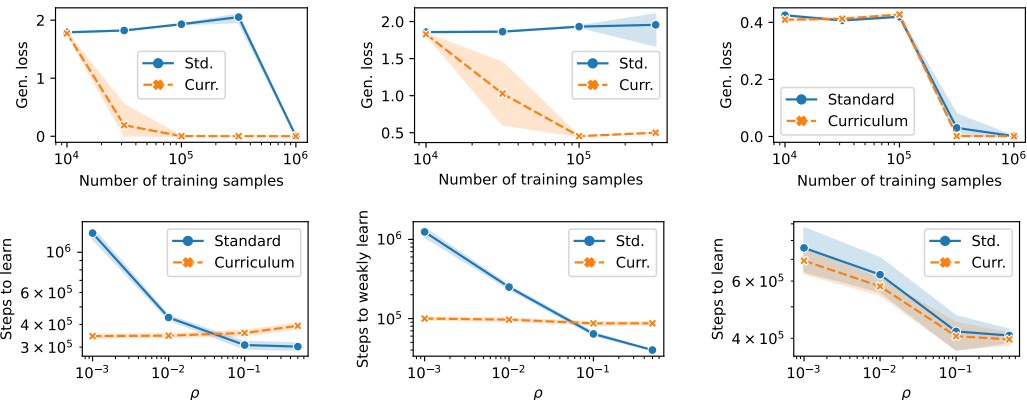

Figure 3: Considering gains in sample complexity (top row) and number of steps (bottom row) beyond parity targets. Plots in the left (corresponding to $f_{\text{left}}(x) = x_1 x_2 x_3 x_4 x_5 + \frac{1}{2} x_1 x_2 x_3 x_4 x_5 x_6$), middle ($f_{\text{middle}}(x) = x_1 x_2 x_3 x_4 x_5 + \frac{1}{2} x_6 \cdots x_{11}$), and right ($f_{\text{right}}(x) = \frac{1}{2} x_1 x_2 + \frac{1}{2} x_1 \cdots x_6$) columns present cases in which curriculum is beneficial, is only beneficial for weak learning, and is not beneficial, respectively.

To analyze the effect of the degree of parity, $k$, we fix $\mu = 0.98, \rho = 0.01$ and the input dimension $d = 100$, and we compare the performance of the model trained with different amounts of training data. The results are portrayed in Figure 2 (bottom-left). One can see that, while without curriculum the parity problem becomes harder to learn as $k$ increases, the complexity with curriculum has mild dependence on $k$ and it allows one to learn all the parities considered with $10^5$ samples. The plot exhibiting the dependence of the number of optimization steps on $k$ is reported in Appendix E.2.

**Beyond Parities.** Finally, we study target functions composed of multiple monomials. We consider three examples in 100-dimensional space: (i) $f_{\text{left}}(x) = x_1 x_2 x_3 x_4 x_5 + \frac{1}{2} x_1 x_2 x_3 x_4 x_5 x_6$, (ii) $f_{\text{middle}}(x) = x_1 x_2 x_3 x_4 x_5 + \frac{1}{2} x_6 \cdots x_{11}$, and (iii) $f_{\text{right}}(x) = \frac{1}{2}(x_1 x_2 + x_1 \cdots x_6)$. We also fix $\mu = 0.98$. In Figure 3, we investigate the gain of curriculum method in sample complexity for $\rho = 0.01$ (top row) and the number of training steps (bottom row) needed to learn. The plots in the bottom row are obtained by training the network with fresh batches. The left, middle, and right column correspond to $f_{\text{left}}, f_{\text{middle}},$ and $f_{\text{right}}$, respectively. Note that curriculum is beneficial for $f_{\text{left}}$, while it hardly helps with $f_{\text{right}}$. Interestingly, for $f_{\text{middle}}$, curriculum is advantageous for weak learning of the function. More specifically, it helps learning $x_1 x_2 x_3 x_4 x_5$ monomial of $f_{\text{middle}}$. Consequently, for the corresponding iterations plot (Figure 3 bottom-middle), we plot the number of iterations needed for getting a loss below $0.26$. This suggests that the curriculum method put forward in this paper mostly helps with learning of the monomial with the lowest degree. We leave improvements of our curriculum method to future work.

## 6 Conclusion

In this paper, we introduced a curriculum strategy for learning parities on mixed distributions. Our approach involves training on sparse samples during the initial phase, which leads to a reduction in the number of required training steps and sample complexity compared to standard training. In the experiments section, we presented some results for certain functions with multiple monomials. Although a complete picture requires further investigation, these results suggest that starting from sparse samples provides advantages in the early stages of training. It would be interesting to investigate what variation of our curriculum strategy fits best the case of multiple monomials, beyond the initial phase. For example, one could explore multiple curriculum phases where training progresses from samples with increasing Hamming weight, or employ a self-paced curriculum that dynamically selects samples during training. Such analyses could offer valuable insights into the broader applicability of our approach. Here we focused mainly on establishing a formal separation in the number of training steps for the canonical case of parities.

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

# A  Proof of Theorem 3

---

**Algorithm 1** Layer-wise curriculum-SGD. Init. scales $\kappa, \Delta > 0$, learning rates $\gamma_1, \gamma_2 > 0$, step counts $T_1, T_2$, input param. $\rho, \mu$, batch size $B$, noise level $\tau$ and gradient range $A$.

---

1: **input:** Data matrix $X \in \{\pm 1\}^{d \times n}$ and label vector $Y \in \{\pm 1\}^n$
2: **initialize:** For all $i \in [N], j \in [d]$, $w_{i,j}^0 = 0, a_i^0 = \kappa, b_i^0 = \Delta d + 1$
3: $X_1 = \{x^s : H(x^s) < d(\frac{1}{2} - \frac{\mu}{4})\}$ and $Y_1 = \{y^s : x^s \in X_1\}$       $\triangleright$ Select sparse inputs
4: **for** $t = 0$ **to** $T_1 - 1$ **: do**
5:      Select unused $x^{s,t} \sim X_1$ and $y^{s,t} \sim Y_1$, $s \in [B]$.
6:      $\forall i \in [N], j \in [d]$:
7:          $\tilde{w}_{ij}^{t+1} = w_{ij}^t - \gamma_1 \left( \frac{1}{B} \sum_{s \in [B]} \partial_{w_{ij}^t} [L(\theta^t, x^{s,t}, y^{s,t})]_A + \text{Unif}[-\tau, \tau] \right)$
8:          $w_{ij}^{t+1} = [\tilde{w}_{ij}^{t+1}]_\Delta$       $\triangleright$ $[.]_\Delta$: proj. on $[-\Delta, \Delta]$
9:          $b_i^{t+1} = b_i^t, a_i^{t+1} = a_i^t$
10: **end for**
11: For all $i \in [N], b_i^{T_1} = -1 + 2(i+1)\Delta, a_i^{T_1} = 0$
12: **for** $t = T_1$ **to** $T_2 - 1$ **: do**
13:      Select unused $x^{s,t} \sim X$ and $y^{s,t} \sim Y$, $s \in [B]$.
14:      $\forall i \in [N]$:
15:          $a_i^{t+1} = a_i^t - \gamma_2 \left( \frac{1}{B} \sum_{s \in [B]} \partial_{a_i^t} [L(\theta^t, x^{s,t}, y^{s,t})]_A + \text{Unif}[-\tau, \tau] \right)$
16:          $w_{ij}^{t+1} = w_{ij}^t, b_i^{t+1} = b_i^t$
17: **end for**

---

In this section, we report the proof of Theorem 3. The proof follows a similar argument used in [TV19, AGJ21, ABAM23]: we decompose the dynamics into a drift and a martingale contribution. Our work differs from the ones above in considering a mixed input distribution and the covariance loss. For the purposes of this section, we consider a 2-layer neural network: $\text{NN}(x; \theta) = \sum_{i=1}^N a_i \sigma(w_i x + b_i)$, with $N = k + 1$ hidden neurons and activation $\sigma := \text{ReLU}$. We consider the following initialization:

$$w_{ij}^0 = 0, \tag{9}$$

$$a_i^0 = \kappa \le \frac{1}{N(2\Delta d + 2)}, \tag{10}$$

$$b_i^0 = d\Delta + 1. \tag{11}$$

We train the neural network using the layer-wise curriculum-SGD algorithm (see Algorithm 1 for pseudo-code). Without loss of generality, we assume that the target parity is $\chi_{[k]} := \prod_{j \in [k]} x_j$. The result for other $k$-parities follows directly since the initialization and the curriculum-SGD algorithm are invariant to permutations of the input neurons. We show the following:

- With a large enough training set, there are enough sparse inputs, which can be identified by computation of the Hamming weight.
- We train the first layer's weights $w_{ij}$ on the sparse data, keeping the $a_i, b_i$ fixed. We first show that with enough sparse samples, the labels' average will concentrate around $\mathbb{E}_{x \sim \mathcal{D}_\mu}[f(x)]$, with high probability. This term is needed in the computation of the covariance loss. Then, we show that after $T_1 = \tilde{O}(d)$ steps, with an appropriate learning rate, with high probability, the target parity will belong to the linear span of the hidden units. This is obtained by showing that the weights corresponding to the first $k$ input coordinates move noticeably, while the ones corresponding to the other $d - k$ coordinates remain small.
- We train the second layer's weights $a_i$ on the full data, keeping the first layer's weights and biases fixed. Established results on the convergence of SGD on convex losses allow us to conclude.

**Bounding the Number of Samples.** In the following lemma, we show that if the training set is large enough, the fraction of sparse inputs is large, and their Hamming weights will concentrate around expectation with high probability as $d$ increases.

**Lemma 1.** *Let $\{x^s\}_{s\in[m]}$ be $m$ i.i.d. inputs sampled from $\mathcal{D}$, with parameters $\rho, \mu$. Let $X_1 := \{x^s : H(x^s) < d(\frac{1}{2} - \frac{\mu}{4})\}$ and let $S_1 := \{x^s : x^s \sim \mathcal{D}_\mu\}$. If $d \leq m \ll \exp(d)$ and $m \geq \frac{2m_1}{\rho}$, for $C^* > 0$,*

   *i) With probability at least $1 - 2d^{-C^*}$, $|S_1| \geq m_1$.*

   *ii) With probability at least $1 - d^{-C^*}$, $S_1 = X_1$.*

*Proof.*      i) For all $s \in [m]$, $\mathbb{P}(x^s \in S^1) = \rho$, thus,

$$\mathbb{P}\Big(\sum_{s\in[m]} \mathbb{1}(x^s \in S_1) \leq m_1\Big) = \mathbb{P}\Big(\sum_{s\in[m]} \mathbb{1}(x^s \in S_1) - m\rho \leq m_1 - m\rho\Big) \tag{12}$$

$$= \mathbb{P}\Big(\sum_{s\in[m]} \mathbb{1}(x^s \in S_1) - m\rho \leq -\frac{m\rho}{2}\Big) \tag{13}$$

$$\overset{(a)}{\leq} \mathbb{P}\Big(m\rho - \sum_{s\in[m]} \mathbb{1}(x^s \in S_1) \geq \sqrt{C^* \log(d)m/2}\Big) \tag{14}$$

$$\overset{(b)}{\leq} 2\exp\left(-2C^* \log(d)\right) \leq 2d^{-C^*}, \tag{15}$$

where in $(a)$ we used that

$$\frac{m\rho}{2} \geq \frac{\sqrt{dm}\rho}{2} \geq \sqrt{C^* \log(d)m/2}, \tag{16}$$

for some constant $C^*$, that depends only on $\rho$, and in $(b)$ we used Hoeffding's inequality.

   ii) It is enough to show that for all $x^s \in S_1$, $H(x^s) < \bar{h}$, for $\bar{h} = d\left(\frac{1}{2} - \frac{\mu}{4}\right)$ and for all $x^s \notin S_1$, $H(x^s) \geq \bar{h}$. Thus, for $x^s \in S_1$,

$$\mathbb{P}\left(H(x^s) \geq \bar{h}\right) = \mathbb{P}\left(H(x^s) - \frac{1-\mu}{2}d \geq \bar{h} - \frac{1-\mu}{2}d\right) \tag{17}$$

$$\leq \exp\left(-\frac{\mu^2 d}{8}\right). \tag{18}$$

Similarly, for $x^s \notin S_1$, $\mathbb{P}\left(H(x^s) < \bar{h}\right) < \exp(-C'd)$, for some $C' > 0$. The result follows from union bound.

□

In the rest of the proof, we will assume that the event in Lemma 1 holds.

## A.1   First Layer Training

For the first training phase, we only use the sparse samples.

**Computing $\bar{y}_1$.**   As a first step, we need to estimate the average of $y$ over the sparse set. To that purpose, we sample one batch of size $B_1$ from $X_1$. We denote by $\bar{y}_1$ such estimate

**Lemma 2.** *Let $\bar{y}_1 = \frac{1}{B_1} \sum_{s\in[B_1]} y^s$. If $B_1 \geq 2C^* \log(d)/\zeta^2$, with probability at least $1 - d^{-C^*}$,*

$$\left|\bar{y}_1 - \mu^k\right| \leq \zeta. \tag{19}$$

*Proof.* By construction, and by Lemma 1, $\bar{y}_1$ is the mean of $B_1$ i.i.d random variables sampled from $\mathcal{D}_\mu$. Thus, by Hoeffding's inequality,

$$\mathbb{P}(|\bar{y}_1 - \mathbb{E}_{x\sim\mathcal{D}_p}[f(x)]| \geq \zeta) \leq 2\exp\left(-\frac{B_1\zeta^2}{2}\right) \leq 2d^{-C^*}. \tag{20}$$

□

We assume that each sample is presented to the network at most once. For instance, this can be done by discarding each sample after it is used, assuming that the number of samples is large enough for covering all training steps. Denoting by $G_{w_{ij}^t}$ the gradient computed at step $t < T_1$ for weight $w_{ij}^t$, the updates of the first layer weights are given by:

$$\tilde{w}_{ij}^{t+1} = w_{ij}^t - \gamma G_{w_{ij}^t} \tag{21}$$

$$w_{ij}^{t+1} = [\tilde{w}_{ij}^{t+1}]_\Delta \tag{22}$$

where by $[.]_\Delta$ we denote the projection of all coordinates that exceed $\Delta$ (resp. $-\Delta$) to $\Delta$ (resp. $-\Delta$). Note that with our assumption on the initialization, $|\operatorname{NN}(x; \theta^t)| < 1$ for all $t < T_1$. Thus, the gradients at each step are given by:

$$G_{w_{ij}^t} = -\frac{1}{B} \sum_{s \in [B]} (y^{s,t} - \bar{y}_1) \cdot \partial_{w_{ij}^t} \operatorname{NN}(x^{s,t}; \theta^t) + Z_{w_{ij}^t}, \tag{23}$$

We decompose the dynamic into a *drift* and a *martingale* contribution:

$$G_{w_{ij}^t} = \underbrace{\bar{G}_{w_{ij}^t}}_{\text{drift}} + \underbrace{(G_{w_{ij}^t} - \bar{G}_{w_{ij}^t})}_{\text{martingale}}, \tag{24}$$

where we denoted by

$$\bar{G}_{w_{ij}^t} = \mathbb{E}_x G_{w_{ij}^t}, \tag{25}$$

the (deterministic) population gradients.

**Drift Contribution.** In the following, we denote by $w_j^t = w_{ij}^t$ for an arbitrary neuron $i \in [N]$. We first bound the drift term.

**Lemma 3.** *Let* $L_\mu := \mu^{k-1} - \mu^{k+1}$. *Let* $\zeta > 0$. *If* $B_1 \geq 2C^* \log(d)/\zeta^2$, *with prob.* $1 - d^{-C^*}$ *for all* $t \leq T_1$,

$$\forall j \in [k]: \quad |\bar{G}_{w_j^t} + \kappa L_\mu| \leq \kappa \mu \zeta; \tag{26}$$

$$\forall j \notin [k]: \quad |\bar{G}_{w_j^t}| \leq \kappa \mu \zeta. \tag{27}$$

*Proof.* With our initialization, for all $t < T_1$,

$$\partial_{w_{ij}^t} \operatorname{NN}(x^t; \theta^t) = \kappa x_j^t. \tag{28}$$

Thus, for all $j \in [k]$,

$$\bar{G}_{w_j^t} = -\kappa \left( \mathbb{E}_x[f(x)x_j] - \bar{y}\mathbb{E}_x[x_j] \right) \tag{29}$$

$$= -\kappa \left( \mu^{k-1} - \bar{y}_1 \mu \right). \tag{30}$$

Applying Lemma 2, with probability $1 - \epsilon'$,

$$|\bar{G}_{w_j^t} + \kappa L_\mu| = |\kappa \mu(\bar{y} - \mu^k)| \leq \kappa \mu \zeta. \tag{31}$$

Similarly, for all $j \notin [k]$,

$$\bar{G}_{w_j^t} = -\kappa \mu(\mu^k - \bar{y}), \tag{32}$$

thus, applying one more time Lemma 2, we get the result. $\qquad \square$

**Martingale Contribution.** We consider $w_j^t = w_{ij}^t$ for an arbitrary neuron $i \in [N]$. For $t \in \mathbb{N}$ and for all $j$, we define the martingale contribution to the dynamics up to time $t$ by

$$M_j^t = \sum_{s=1}^t G_{w_j^s} - \bar{G}_{w_j^s}. \tag{33}$$

We show that for appropriate time horizon and learning rate, the martingale contribution stays small.

**Lemma 4** (Lemma 4 in [ABAM23]). *Fix $T \leq C_0 d \log(d)^{C_0}$, for some $C_0 > 0$. For all $C^*$, there exists $C$ that depends on $C_0, C^*$ such that if:*

$$\gamma_1^2 T \leq \frac{1}{C(4\kappa + \tau)^2 \log(d)^2 d}, \tag{34}$$

*with probability at least $1 - d^{-C^*}$:*

$$\max_{0 < t < T} \max_{j \in [d]} |\gamma_1 M_j^t| \leq \frac{1}{\sqrt{d \log(d)}}. \tag{35}$$

*Proof.* Note that in our setting (Boolean inputs, ReLU activation) we have that for all $t < T_1$

$$|M_j^t - M_j^{t-1}| \leq |G_{w_j^t}| + |\bar{G}_{w_j^t}| \leq 4\kappa + \tau. \tag{36}$$

Thus, by Azuma-Hoeffding inequality:

$$\mathbb{P}\left(|\gamma_1 M_j^t| \geq \epsilon\right) \leq 2 \exp\left(-\frac{\epsilon^2}{2T\gamma_1^2(4\kappa + \tau)^2}\right). \tag{37}$$

The result follows by choosing $\epsilon = 1/\sqrt{d \log(d)}$ and by union bound on all $t \leq C_0 d \log(d)^{C_0}$ and $j \in [d]$. $\qquad \square$

**Bounding the Contributions to the Dynamics.**    We define the following stopping times:

$$\tau_j^\Delta := \inf\{t \geq 0 : |w_j^{t+1}| \geq \Delta - \gamma_1(4\kappa + \tau)\} \tag{38}$$

$$\tau^\Delta = \sup_{j \in [k]} \tau_j^\Delta. \tag{39}$$

We show that for $t \geq \tau_j^\Delta$, $w_j^t$ stays close to $\Delta$.

**Lemma 5.** *Fix $T \leq C_0 d \log(d)^{C_0}$, for some $C_0 > 0$. Let $\zeta > 0$ and let $B_1 \geq 2C^* \log(d)/\zeta^2$. For all $C^*$, there exists $C$ such that if $\gamma_1$ satisfies (34) and if $\zeta \leq \frac{1}{\sqrt{d \log(d)^{C_0}}}$, with probability $1 - 3d^{-C^*}$ for all $j \in [d]$:*

$$\inf_{\tau_j^\Delta < t < T} w_j^t \geq \Delta - \frac{3 + \sqrt{C_0}\mu}{\sqrt{d \log(d)}}. \tag{40}$$

*Proof.* For all $j$, let $s = \sup\{t' \leq t : w_j^{t'} \geq \Delta - \gamma_1(4\kappa + \tau)\}$, then with probability $1 - 3d^{-C^*}$ for all $j \in [d]$:

$$w_j^t = w_j^s - \gamma_1 \sum_{l=s}^{t-1} \bar{G}_{w_j^l} + \gamma_1(M_j^{t-1} - M_j^s) \tag{41}$$

$$\overset{(a)}{\geq} \Delta - \gamma_1(4\kappa + \tau) - \gamma_1 t\kappa\mu\zeta - \frac{2}{\sqrt{d \log(d)}} \tag{42}$$

$$\overset{(b)}{\geq} \Delta - \frac{3}{\sqrt{d \log(d)}} - \frac{\sqrt{t}\zeta\mu}{\sqrt{d \log(d)}}, \tag{43}$$

$$\overset{(c)}{\geq} \Delta - \frac{3}{\sqrt{d \log(d)}} - \frac{\sqrt{C_0}\mu}{\sqrt{d \log(d)}}, \tag{44}$$

where in $(a)$ we used that by Lemma 3 with prob. $1 - 2d^{-C^*}$, $\bar{G}_{w_j^l} < -\kappa L_\mu + \kappa\mu\zeta < \kappa\mu\zeta$ for all $l \in [t]$ and Lemma 4, in $(b)$ we used that by (34), $\gamma_1(4\kappa + \tau) \leq \gamma_1(4\kappa + \tau)\sqrt{t} \leq 1/\sqrt{d \log(d)}$, and in $(c)$ we used the assumptions on $T$ and $\zeta$. Thus, the result holds. $\qquad \square$

We will now establish a high-probability bound on the time that it takes for all weights associated with coordinates in the support of the target parity to approach proximity with $\Delta$.

**Lemma 6.** *Let $\zeta > 0$ and let $B_1 \geq 2C^* \log(d)/\zeta^2$. For all $C^*$, there exists $C$ such that if $\gamma_1$ satisfies (34), $\zeta \leq \frac{1}{\sqrt{d \log(d)^{C_0}}}$ and $d$ is large enough, such that $d \log(d) \geq \frac{(\sqrt{C_0}\mu+1)^2}{\Delta^2}$, with probability $1 - 3d^{-C^*}$:*

$$\tau^\Delta \leq \frac{2\Delta}{\gamma_1 \kappa L_\mu}. \tag{45}$$

*Proof.* Let $T \leq C_0 d \log(d)^{C_0}$, for some $C_0 > 0$. Let $\alpha_t = \min\{w_j^t : j \in [k], t \leq \tau_j^\Delta, t \leq T\}$. Then, with probability $1 - 3d^{-C^*}$:

$$\alpha_t \geq -\gamma_1 \sum_{s=0}^{t-1} \bar{G}_{\alpha_s} - \gamma_1 |M_j^t| \tag{46}$$

$$\geq -\gamma_1 t(-\kappa L_\mu + \kappa\mu\zeta) + \frac{1}{\sqrt{d \log(d)}} \tag{47}$$

$$\overset{(a)}{\geq} \gamma_1 t\kappa L_\mu - \frac{\sqrt{t}\zeta\mu}{\sqrt{d \log(d)}} - \frac{1}{\sqrt{d \log(d)}} \tag{48}$$

$$\overset{(b)}{\geq} \gamma_1 t\kappa L_\mu - \frac{\sqrt{C_0}\mu + 1}{\sqrt{d \log(d)}}, \tag{49}$$

where in $(a)$ we used that by (34), $\sqrt{t}\gamma_1\kappa \leq 1/\sqrt{d\log(d)}$, and in $(b)$ we used the assumptions on $T$ and $\zeta$. Thus if $t \geq \frac{2\Delta}{\gamma_1 \kappa L_\mu}$ and $d$ is large enough, such that $\frac{\sqrt{C_0}\mu+1}{\sqrt{d\log(d)}} < \Delta$, we have

$$\alpha_t \geq 2\Delta - \Delta > \Delta - \gamma_1(4\kappa + \tau). \tag{50}$$

Thus, the result holds.

$\square$

As our last lemma, we show that during the same time horizon, all coordinates that are not in the support of the parity will stay close to zero.

**Lemma 7.** *Fix $T \leq C_0 d \log(d)^{C_0}$, for some $C_0 > 0$. Let $\zeta > 0$ and let $B_1 \geq 2C^* \log(d)/\zeta^2$. For all $C^*$, there exists $C$ such that if $\gamma_1$ satisfies (34), $\zeta \leq \frac{1}{\sqrt{d \log(d)^{C_0}}}$ and $d$ is large enough, such that $d \log(d) \geq \frac{(\sqrt{C_0}\mu+1)^2}{\Delta^2}$, with probability $1 - 3d^{-C^*}$:*

$$\sup_{0 \leq t \leq T} \sup_{j \in \{k+1,\dots,d\}} |w_j^t| \leq \frac{\sqrt{C_0}\mu + 1}{\sqrt{d \log(d)}}. \tag{51}$$

*Proof.* Let $|\beta_t| = \max\{|w_j^t| : j \in [k+1,\dots,d], t \leq T\}$. Thus, with probability $1 - 3d^{-C^*}$:

$$|\beta_t| = \left| -\gamma_1 \sum_{s=0}^{t-1} \bar{G}_{\beta_s} - \gamma_1 M_j^t \right| \tag{52}$$

$$\leq \gamma_1 \sum_{s=0}^{t-1} |\bar{G}_{\beta_s}| + \gamma_1 |M_j^t| \tag{53}$$

$$\leq \gamma_1 t\kappa\mu\zeta + \frac{1}{\sqrt{d \log(d)}} \tag{54}$$

$$\leq \frac{\sqrt{t}\zeta\mu + 1}{\sqrt{d \log(d)}} \tag{55}$$

$$\leq \frac{\sqrt{C_0}\mu + 1}{\sqrt{d \log(d)}}. \tag{56}$$

$\square$

Thus, combining Lemmas 5, 6 and 7 and by union bound on $i \in [N]$, we get that for all $C^*$, there exist $C_1, C_2$ such that if $T_1$ and $\gamma_1$ are such that

$$\gamma_1^2 T_1 \leq \frac{1}{C(4\kappa + \tau)^2 \log(d)^2 d}, \qquad \frac{2\Delta}{\gamma_1 \kappa L_\mu} \leq T_1 \leq C_0 d \log(d)^{C_0}. \tag{57}$$

for some $C_0, C$, with prob. $1 - 3Nd^{-C^*}$, for all $i \in [N]$,

$$\text{For all } j \in [k]: \qquad |w_{ij}^{T_1} - \Delta| \leq \frac{C_1}{\sqrt{d \log(d)}}; \tag{58}$$

$$\text{For all } j \notin [k]: \qquad |w_{ij}^{T_1}| \leq \frac{C_2}{\sqrt{d \log(d)}}. \tag{59}$$

For $\tau \leq \kappa$, we can choose

$$\gamma_1 = \frac{L_\mu}{C' \Delta \kappa d \log(d)^2}, \qquad T_1 = \frac{2C' \Delta^2 d \log(d)^2}{L_\mu^2}, \tag{60}$$

where $C' = C/50$, and the result follows.

## A.2   Second Layer Training

As mentioned, the second phase of training consists of training the second layer's weights $a_i$, keeping $w_{ij}$ and $b_i$ fixed. We do not train the biases, and directly assign $b_i^{T_1}$ to be well spread in the interval $[\Delta(2 - k), \Delta(2 + k)]$, so that the target parity belongs to the linear span of the hidden units at time $T_1$. We believe that one could extend the argument to cover scenarios where biases are trained, but this would require further technical work. To train the second layer, we use samples drawn from the entire dataset. Again, we take a large enough dataset and we assume that each sample is used at most once. In the following, we assume that the events in (58)-(59) hold. Below, we show that there exist $a_i^*$ such that $\sum_{i=1}^{N} a_i^* \sigma(w_i^{T_1} x + b_i)$ achieves small expected loss. For simplicity, we restrict our attention to $k$ even, but an analogous Lemma to the below can be written for $k$ odd. For a function $f$, an estimator $\hat{f}$ and input distribution $\mathcal{D}$, let us define

$$L_{\text{cov}}(\hat{f}, f, \mathcal{D}) := \mathbb{E}_{x \sim \mathcal{D}}\left(1 - f(x)\mathbb{E}_{x \sim \mathcal{D}}f(x) - \hat{f}(x)(f(x) - \mathbb{E}_{x \sim \mathcal{D}}f(x))\right)_+. \tag{61}$$

**Lemma 8.** *Assume $k$ is an even integer. Let $N = k + 1$ and for $i \in \{0, ..., k\}$ let $b_i = -\Delta k + 2(i + 1)\Delta$. For all $\epsilon > 0$, there exist $\Delta > 0, D > \max\{C_1, C_2\}$ such that if $\rho < (\frac{\Delta}{2N} - D)\frac{\sqrt{\log(d)}}{\sqrt{d}\mu}$, and if $d$ is large enough:*

$$\min_{a: \|a\|_\infty \leq 4/\Delta} L_{\text{cov}}(f^*, \chi_{[k]}, \mathcal{D}) < \epsilon/2. \tag{62}$$

*Proof.* We follow a similar proof scheme as in [BEG$^+$22], and choose $a_i^*$ such that

$$a_i^* = (-1)^i \frac{4}{\Delta}, \qquad \forall i \in \{0, 1, \ldots, k - 2\}, \tag{63}$$

$$a_{k-1}^* = (-1)^{k-1} \frac{3}{\Delta}, \tag{64}$$

$$a_k^* = (-1)^k \frac{1}{\Delta}. \tag{65}$$

We denote $f^*(x) = \sum_{i=0}^{k} a_i^* \sigma(w_i^{T_1} x + b_i)$. One can verify that $2\chi_{[k]}(x) = \sum_{i=0}^{k} a_i^* \sigma(\Delta \sum_{j=1}^{k} x_j + b_i)$. For all $i \in [N]$, let us denote $\hat{w}_{ij} = w_{ij}^{T_1} - \Delta$ for all $j \in [k]$ and $\hat{w}_{ij} = w_{ij}^{T_1}$ for all $j \notin [k]$. Thus, $\mathbb{E}_{x \sim \mathcal{D}}[\sum_{j=1}^{d} \hat{w}_{ij} x_j] = \rho \mu \sum_{j=1}^{d} \hat{w}_{ij}$ conditioned on the event in (58)-(59), $\|\hat{w}_i\|_1 \leq \frac{C_3 \sqrt{d}}{\sqrt{\log(d)}}$ for all

$i \in [N]$, where $C_3 = \max\{C_1, C_2\}$. Let $\delta = \frac{\Delta}{8N}$. Then,

$$\mathbb{P}_{x \sim \mathcal{D}}\left(\left|\sum_{j=1}^{d} \hat{w}_{ij} x_j\right| > \delta\right) \leq \mathbb{P}_{x \sim \mathcal{D}}\left(\left|\sum_{j=1}^{d} \hat{w}_{ij} x_j - \rho\mu\|\hat{w}_i\|_1\right| > \delta - \rho\mu\|\hat{w}_i\|_1\right) \tag{66}$$

$$\overset{(a)}{\leq} \mathbb{P}_{x \sim \mathcal{D}}\left(\left|\sum_{j=1}^{d} \hat{w}_{ij} x_j - \rho\mu\|\hat{w}_i\|_1\right| > D\right) \tag{67}$$

$$\overset{(b)}{\leq} 2\exp\left(-\frac{D^2}{2C_3^2}\log(d)\right) \leq 2d^{-\frac{D^2}{2C_3^2}}, \tag{68}$$

where in $(a)$ we used the assumption on $\rho$ and in $(b)$ we used Hoeffding's inequality and the fact that $\sum_{j=1}^{d}(2\hat{w}_{ij})^2 \leq \frac{4C_3^2}{\log(d)}$. Thus, with probability $1 - 2Nd^{-\frac{D^2}{2C_3^2}}$ over $x$ for all $i \in \{0, ..., k\}$:

$$\left|\sigma(w_i^{T_1} x + b_i) - \sigma(\Delta \sum_{j=1}^{k} x_j + b_i)\right| \leq \left|\sum_{j=1}^{d} \hat{w}_{ij} x_j\right| \leq \delta. \tag{69}$$

Consequently,

$$|2\chi_{[k]}(x) - f^*(x)| = \left|\sum_{i=0}^{k} a_i^*\left[\sigma(w_i^{T_1} x + b_i) - \sigma(\Delta \sum_{j=1}^{k} x_j + b_i)\right]\right| \tag{70}$$

$$\leq \sum_{i=0}^{k} |a_i^*|\delta \leq \frac{4N}{\Delta}\delta = 1/2 \tag{71}$$

Thus, for $d$ large enough, with probability $1 - 2Nd^{-\frac{D^2}{2C_3^2}}$ we have $f^*(x)\chi_{[k]}(x) \geq 1$. Moreover, for all $x$,

$$|f^*(x)| \leq N\|a^*\|_\infty \left(2\Delta k + \frac{\sqrt{d}C_3}{\sqrt{\log(d)}}\right) \tag{72}$$

$$\leq \frac{\sqrt{d}}{\sqrt{\log(d)}} \frac{4N(C_3 + 2\Delta k)}{\Delta}. \tag{73}$$

Then,

$$L_{\text{cov}}(f^*, \chi_{[k]}, \mathcal{D}) = \mathbb{E}_{x \sim \mathcal{D}}\left((1 - \chi_{[k]}(x)\mathbb{E}_x[\chi_{[k]}(x)])(1 - \chi_{[k]}(x)f^*(x))\right)_+ \tag{74}$$

$$= \mathbb{E}_{x \sim \mathcal{D}}(1 - \chi_{[k]}(x)\rho\mu^k) \cdot (1 - \chi_{[k]}(x)f^*(x)) \cdot \mathbb{1}(\chi_{[k]}(x)f^*(x) < 1) \tag{75}$$

$$\leq \frac{9N^2(C_3 + 2\Delta k)(1 + \rho\mu^k)}{\Delta\sqrt{\log(d)}} \cdot d^{-\frac{D^2}{2C_3^2} + \frac{1}{2}} \tag{76}$$

$$< \epsilon/2, \tag{77}$$

for $d$ large enough. $\qquad\square$

To conclude, we apply an established result from [SSBD14] on the convergence of SGD on convex losses.

**Theorem 6** ([SSBD14]). *Let $\mathcal{L}$ be a convex function and let $a^* \in \arg\min_{\|a\|_2 \leq \mathcal{B}} \mathcal{L}(a)$, for some $\mathcal{B} > 0$. For all $t$, let $\alpha^t$ be such that $\mathbb{E}[\alpha^t \mid a^t] = -\nabla_{a^t}\mathcal{L}(a^t)$ and assume $\|\alpha^t\|_2 \leq \xi$ for some $\xi > 0$. If $a^{(0)} = 0$ and for all $t \in [T]$ $a^{t+1} = a^t + \gamma\alpha^t$, with $\gamma = \frac{\mathcal{B}}{\xi\sqrt{T}}$, then*

$$\frac{1}{T}\sum_{t=1}^{T} \mathcal{L}(a^t) \leq \mathcal{L}(a^*) + \frac{\mathcal{B}\xi}{\sqrt{T}}. \tag{78}$$

We refer to [SSBD14] for a proof. Let us take $\mathcal{L}(a) := L_{\text{cov}}((a, w^{T_1}, b^{T_1}), \chi_{[k]}, \mathcal{D})$. Then, $\mathcal{L}$ is convex in $a$ and for all $t \in [T_2]$,

$$\alpha^t = -\frac{1}{B} \sum_{s \in [B]} \nabla_{a^t} L_{\text{cov}}((a^t, w^{T_1}, b^{T_1}), \chi_{[k]}, x^{s,t}) + Z_{a^t}, \tag{79}$$

and, recalling $\sigma := \text{ReLU}$, we have

$$\|\alpha^t\|_\infty \leq 2 \sup_x |\sigma(w_i^{T_1} x + b_i^{T_1})| + \tau \tag{80}$$

$$\leq 2 \left( 2\Delta k + \frac{\sqrt{d} C_3}{\sqrt{\log(d)}} \right) + \tau \tag{81}$$

$$\leq 2 \left( 2\Delta k + C_3 + \tau \right) \sqrt{\frac{d}{\log(d)}}, \tag{82}$$

and $\|\alpha^t\|_2 \leq \sqrt{N} \|\alpha^t\|_\infty$. Thus, if we choose

- $\mathcal{B} = \frac{4\sqrt{N}}{\Delta}$
- $\xi = 2(C_3 + 2\Delta k + \tau) \sqrt{\frac{dN}{\log(d)}}$
- $T_2 = \frac{256(C_3 + 2\Delta k + \tau)^2 N^2 d}{\epsilon^2 \Delta^2 \log(d)}$
- $\gamma_2 = \frac{\epsilon \log(d)}{8(C_3 + 2\Delta k + \tau)^2 dN}$

we obtain

$$\min_{t \in [T_1 + T_2]} L_{\text{cov}} \left( \theta^t, \chi_{[k]}, x \right) \leq \frac{\epsilon}{2} + \frac{\epsilon}{2} = \epsilon. \tag{83}$$

The result follows under the assumption that the training process is terminated once the training loss falls below $\epsilon$, thus guaranteeing that $L_{\text{cov}} \left( \theta^{T_1 + T_2}, \chi_{[k]}, x \right) \leq \epsilon$.

## B  Proof of Theorem 4

**Theorem 7** (Theorem 4, restatement). *Let $d, N$ be two positive integers. Let $\text{NN}(x; \theta) = \sum_{i=1}^N a_i \sigma(w_i x + b_i)$ be a 2-layer fully connected network with activation $\sigma(y) := \text{Ramp}(y)$ (as defined in (84)) and $N \geq (d+1)(d-k+1) \log((d+1)(d-k+1)/\delta)$. Consider training $\text{NN}(x; \theta)$ with layerwise curriculum-SGD on the hinge loss with batch size $B \geq (4\zeta^2 N^2)^{-1} \log(\frac{Nd+N}{\delta})$, noise level $\tau \leq \zeta \log \left( \frac{Nd+N}{\delta} \right)^{-1/2}$, with $\zeta \leq \frac{\epsilon \mu^k}{24(d+1)^2(d-k+1)^2 N}$ and $\mu = \sqrt{1 - \frac{1}{2(d-k)}}$. Then, there exists an initialization and a learning rate schedule such that after $T_1 = 1$ step on the sparse data and $T_2 = \frac{64}{\epsilon^2}(d-k+1)^3(d+1)N(1+\tau)^2$ steps on the full data, with probability $1 - 3\delta$, the trained network has generalization error at most $\epsilon$.*

This proof is similar to the proof of Theorem 3 in [CM23], with the following two modifications: 1) the use of a mixed input distribution in the second part of training, and, 2) the addition of uniform gradient noise in the iterates.

### B.1  Proof Setup

We consider a 2-layer neural network, defined as $\text{NN}(x; \theta) = \sum_{i=1}^N a_i \sigma(w_i x + b_i)$, where $N \geq (d+1)(d-k+1) \log((d+1)(d-k+1)/\delta)$ is the number of hidden units, $\theta = (a, b, w)$ and $\sigma := \text{Ramp}$ denotes the activation defined as:

$$\text{Ramp}(x) = \begin{cases} 0 & x \leq 0, \\ x & 0 < x \leq 1, \\ 1 & x > 1 \end{cases} . \tag{84}$$

Similarly as before, without loss of generality, we assume that the labels are generated by $\chi_{[k]}(x) := \prod_{i=1}^k x_i$. Our proof scheme is the following:

1. We train only the first layer of the network for one step on data $(x, \chi_{[k]}(x))$ with $x \sim \mathcal{D}_\mu$, with $\mu := 2p - 1 = \sqrt{1 - \frac{1}{2(d-k)}}$;

2. We show that after one step of training on such biased distribution, the target parity belongs to the linear span of the hidden units of the network;

3. We train only the second layer of the network on $(x, \chi_{[k]}(x))$ with $x \sim \mathcal{D}$, until convergence;

4. We use established results on the convergence of SGD on convex losses to conclude.

We train our network with noisy-GD on the hinge loss. Specifically, we apply the following updates, for all $t \in \{0, 1, \dots, T - 1\}$:

$$
w_{i,j}^{t+1} = w_{i,j}^t - \gamma_t \left( \frac{1}{B} \sum_{s=1}^B \nabla_{w_{i,j}^t} L(\theta^t, \chi_{[k]}, x_s^t) + Z_{w_{i,j}^t} \right),
$$

$$
a_i^{t+1} = a_i^t - \eta_t \left( \frac{1}{B} \sum_{s=1}^B \nabla_{a_i^t} L(\theta^t, \chi_{[k]}, x_s^t) + Z_{a_i^t} \right) + c_t, \tag{85}
$$

$$
b_i^{t+1} = \lambda_t \left( b_i^t + \psi_t \frac{1}{B} \sum_{s=1}^B \nabla_{b_i^t} L(\theta^t, \chi_{[k]}, x_s^t) + Z_{b_i^t} \right) + d_t,
$$

where $L(\theta^t, \chi_{[k]}, x) = \max\{0, 1 - \chi_{[k]}(x) \, \mathrm{NN}(x; \theta^t)\}$ and $Z_{w_{i,j}^t}, Z_{a_i^t}$ and $Z_{b_i^t}$ are i.i.d. $\mathrm{Unif}[-\tau, \tau]$ for some $\tau > 0$. Following the proof strategy introduced above, we set $\mathcal{D}^0 = \mathcal{D}_\mu$, where $\mu = \sqrt{1 - \frac{1}{2(d-k)}}$, and $\mathcal{D}^t = \mathcal{D}$, for all $t \geq 1$. We set the parameters of (85) to:

$$
\gamma_0 = \mu^{-(k-1)} 2N, \qquad\qquad \gamma_t = 0 \qquad \forall t \geq 1, \tag{86}
$$

$$
\eta_0 = 0, \qquad\qquad \eta_t = \frac{\epsilon}{2N(1+\tau)^2} \qquad \forall t \geq 1, \tag{87}
$$

$$
\psi_0 = \frac{N}{\mu^k}, \qquad\qquad \psi_t = 0 \qquad \forall t \geq 1, \tag{88}
$$

$$
c_0 = -\frac{1}{2N}, \qquad\qquad c_t = 0 \qquad \forall t \geq 1, \tag{89}
$$

$$
\lambda_0 = (d+1), \qquad\qquad \lambda_t = 1 \qquad \forall t \geq 1, \tag{90}
$$

$$
d_0 = 0, \qquad\qquad d_t = 0 \qquad \forall t \geq 1, \tag{91}
$$

and we consider the following initialization scheme:

$$
w_{i,j}^0 = 0 \qquad \forall i \in [N], j \in [d];
$$

$$
a_i^0 = \frac{1}{2N} \qquad \forall i \in [N]; \tag{92}
$$

$$
b_i^0 \sim \mathrm{Unif}\left\{ \frac{b_{lm}}{d+1} + \frac{1}{2} : l \in \{0, ..., d\}, m \in \{-1, ..., d-k\} \right\},
$$

where we define

$$
b_{lm} := -d + 2l - \frac{1}{2} + \frac{m+1}{d-k}. \tag{93}
$$

Note that such initialization is invariant to permutations of the input neurons.

## B.2 First Step: Recovering the Support

As mentioned above, we train our network for one step on $(x, \chi_{[k]}(x))$ with $x \sim \mathcal{D}_\mu$.

**Population Gradient at Initialization.** Let us compute the population gradient at initialization. Since we set $\xi_0 = 0$, we do not need to compute the initial gradient for $a$. Note that at initialization

$|\operatorname{NN}(x; \theta^0)| < 1$. Thus, the initial population gradients are given by

$$\forall j \in [k], i \in [N] \qquad \bar{G}_{w_{i,j}} = -a_i \mathbb{E}_{x \sim \mathcal{D}_\mu} \left[ \prod_{l \in [k] \backslash j} x_l \cdot \mathbb{1}(\langle w_i, x \rangle + b_i \in [0,1]) \right] \tag{94}$$

$$\forall j \notin [k], i \in [N] \qquad \bar{G}_{w_{i,j}} = -a_i \mathbb{E}_{x \sim \mathcal{D}_\mu} \left[ \prod_{l \in [k] \cup j} x_l \cdot \mathbb{1}(\langle w_i, x \rangle + b_i \in [0,1]) \right] \tag{95}$$

$$\forall i \in [N] \qquad \bar{G}_{b_i} = -a_i \mathbb{E}_{x \sim \mathcal{D}_\mu} \left[ \prod_{l \in [k]} x_l \cdot \mathbb{1}(\langle w_i, x \rangle + b_i \in [0,1]) \right] \tag{96}$$

**Lemma 9.** *Initialize $a, b, w$ according to (92). Then,*

$$\forall j \in [k], \qquad \bar{G}_{w_{i,j}} = -\frac{\mu^{k-1}}{2N}; \tag{97}$$

$$\forall j \notin [k], \qquad \bar{G}_{w_{i,j}} = -\frac{\mu^{k+1}}{2N}; \tag{98}$$

$$\bar{G}_{b_i} = -\frac{\mu^k}{2N}. \tag{99}$$

*Proof.* If we initialize according to (92), we have $\langle w_i, x \rangle + b_i \in [0,1]$ for all $i$. The results holds since $\mathbb{E}_{x \sim \mathcal{D}_\mu}[\chi_S(x)] = \mu^{|S|}$. □

### Effective Gradient at Initialization.
**Lemma 10** (Noisy-GD). *Let*

$$\hat{G}_{w_{i,j}} := \frac{1}{B} \sum_{s=1}^{B} \nabla_{w_{i,j}^0} L(\theta^0, \chi_{[k]}, x^{s,0}) + Z_{w_{i,j}}, \tag{100}$$

$$\hat{G}_{b_i} := \frac{1}{B} \sum_{s=1}^{B} \nabla_{b_i^0} L(\theta^0, \chi_{[k]}, x^{s,0}) + Z_{b_i}, \tag{101}$$

*be the effective gradients at initialization, where $G_{w_{i,j}}, G_{b_i}$ are the population gradients at initialization and $Z_{w_{i,j}}, Z_{b_i}$ are i.i.d. $\operatorname{Unif}[-\tau, \tau]$. If $B \geq (4\zeta^2 N^2)^{-1} \log(\frac{Nd+N}{\delta})$ and $\tau \leq \zeta \log \left( \frac{Nd+N}{\delta} \right)^{-1/2}$, with probability $1 - 2\delta$:*

$$\|\hat{G}_{w_{i,j}} - \bar{G}_{w_{i,j}}\|_\infty \leq \zeta, \tag{102}$$
$$\|\hat{G}_{b_i} - \bar{G}_{b_i}\|_\infty \leq \zeta. \tag{103}$$

*Proof.* By Hoeffding's inequality

$$\mathbb{P}\left( |\hat{G}_{w_{i,j}} - \bar{G}_{w_{i,j}}| \geq \zeta \right) \leq 2 \exp\left( -\frac{2\zeta^2}{(4N^2 B)^{-1} + \tau^2} \right) \leq \frac{2\delta}{Nd+N}, \tag{104}$$

$$\mathbb{P}\left( |\hat{G}_{b_i} - \bar{G}_{b_i}| \geq \zeta \right) \leq 2 \exp\left( -\frac{2\zeta^2}{(4N^2 B)^{-1} + \tau^2} \right) \leq \frac{2\delta}{Nd+N}. \tag{105}$$

The result follows by union bound. □

**Lemma 11.** *Let*

$$w_{i,j}^1 = w_{i,j}^0 - \gamma_0 \hat{G}_{w_{i,j}} \tag{106}$$

$$b_i^1 = \lambda_0 \left( b_i^0 - \psi_0 \hat{G}_{b_i} \right) \tag{107}$$

$$\tag{108}$$

*If $B \geq (4\zeta^2 N^2)^{-1} \log(\frac{Nd+N}{\delta})$ and $\tau \leq \zeta \log \left( \frac{Nd+N}{\delta} \right)^{-1/2}$, with probability $1 - 2\delta$,*

*i) For all $j \in [k]$, $i \in [N]$, $|w_{i,j}^1 - 1| \leq \frac{2N\zeta}{\mu^{k-1}}$;*

*ii) For all $j \notin [k]$, $|w_{i,j}^1 - (1 - \frac{1}{2(d-k)})| \leq \frac{2N\zeta}{\mu^{k-1}}$;*

*iii) For all $i \in [N]$, $|b_i^1 - (d+1)(b_i^0 - \frac{1}{2})| \leq \frac{N(d+1)\zeta}{\mu^k}$.*

*Proof.* We apply Lemma 11:

i) For all $j \in [k]$, $i \in [N]$, $|\hat{w}_{i,j}^1 - 1| = \gamma_0|\hat{G}_{w_{i,j}} - G_{w_{i,j}}| \leq \frac{2N\zeta}{\mu^{k-1}}$;

ii) For all $j \notin [k]$, $i \in [N]$, $|\hat{w}_{i,j}^1 - (1 - \frac{1}{2(d-k)})| = \gamma_0|\hat{G}_{w_{i,j}} - G_{w_{i,j}}| \leq \frac{2N\zeta}{\mu^{k-1}}$;

iii) For all $i \in [N]$,

$$|\hat{b}_i^1 - (d+1)(b_i^0 - \frac{1}{2})| = |\lambda_0(b_i^0 + \psi_0\hat{G}_{b_i}) - \lambda_0(b_i^0 + \psi_0 G_{b_i})| \tag{109}$$

$$\leq |\lambda_0| \cdot |\psi_0| \cdot |\hat{G}_{b_i} - G_{b_i}| \tag{110}$$

$$\leq \frac{N(d+1)\zeta}{\mu^k}. \tag{111}$$

$\square$

**Lemma 12.** *If $N \geq (d+1)(d-k+1)\log((d+1)(d-k+1)/\delta)$, then with probability $1 - \delta$, for all $l \in \{0, ..., d\}$, and for all $m \in \{-1, ..., d-k\}$ there exists $i$ such that $b_i^0 = \frac{b_{lm}}{d+1} + \frac{1}{2}$.*

*Proof.* The probability that there exist $l, m$ such that the above does not hold is

$$\left(1 - \frac{1}{(d+1)(d-k+1)}\right)^N \leq \exp\left(-\frac{N}{(d+1)(d-k+1)}\right) \leq \frac{\delta}{(d+1)(d-k+1)}. \tag{112}$$

The result follows by union bound. $\square$

**Lemma 13.** *Let $\sigma_{lm}(x) = \text{Ramp}\left(\sum_{j=1}^d x_j - \frac{1}{2(d-k)}\sum_{j>k} x_j + b_{lm}\right)$, with $b_{lm}$ given in (93). If $B \geq (4\zeta^2 N^2)^{-1}\log(\frac{Nd+N}{\delta})$, $\tau \leq \zeta\log\left(\frac{Nd+N}{\delta}\right)^{-1/2}$ and $N \geq (d+1)(d-k+1)\log((d+1)(d-k+1)/\delta)$, with probability $1 - 3\delta$, for all $l, m$ there exists $i$ such that*

$$\left|\sigma_{lm}(x) - \text{Ramp}\left(\sum_{j=1}^d \hat{w}_{i,j}^1 x_j + \hat{b}_i^1\right)\right| \leq 3N(d+1)\zeta\mu^{-k}. \tag{113}$$

*Proof.* By Lemma 12, with probability $1 - \delta$, for all $l, m$ there exists $i$ such that $b_i^{(0)} = \frac{b_{lm}}{d+1} + \frac{1}{2}$. For ease of notation, we replace indices $i \mapsto (lm)$, and denote $\hat{\sigma}_{lm}(x) = \text{Ramp}\left(\sum_{j=1}^d w_{lm,j}^1 x_j + b_{lm}^1\right)$. Then, by Lemma 11 with probability $1 - 2\delta$,

$$|\sigma_{lm}(x) - \hat{\sigma}_{lm}(x)| \leq \left|\sum_{j=1}^k (w_{lm,j}^1 - 1)x_j + \sum_{j=k+1}^d \left(w_{lm,j}^1 - \left(1 - \frac{1}{2(d-k)}\right)\right)x_j + b_{lm}^1 - b_{lm}\right|$$

$$\leq k2N\zeta\mu^{-(k-1)} + (d-k)2N\zeta\mu^{-(k-1)} + N(d+1)\zeta\mu^{-k}$$

$$\leq 3N(d+1)\zeta\mu^{-k}.$$

$\square$

We make use of Lemma 6 from [CM23], for which we rewrite the statement here for completeness.

**Lemma 14** (Lemma 6 in [CM23]). *There exists $a^*$ with $\|a^*\|_\infty \le 4(d-k)$ such that*

$$\sum_{l=0}^{d} \sum_{m=-1}^{d-k} a_{lm}^* \sigma_{lm}(x) = \chi_{[k]}(x). \tag{114}$$

**Lemma 15.** *Let $f^*(x) = \sum_{l,m} a_{lm}^* \sigma_{lm}(x)$ and let $\hat{f}(x) = \sum_{l,m} a_{lm}^* \hat{\sigma}_{lm}(x)$, with $\sigma_{lm}, \hat{\sigma}_{lm}$ defined in Lemma 13 and $a^*$ defined in Lemma 14. If $B \ge (4\zeta^2 N^2)^{-1} \log(\frac{Nd+N}{\delta})$, $\tau \le \zeta \log\left(\frac{Nd+N}{\delta}\right)^{-1/2}$ and $N \ge (d+1)(d-k+1)\log((d+1)(d-k+1)/\delta)$, with probability $1 - 3\delta$ for all $x$,*

$$L(\hat{f}, f^*, x) \le (d+1)^2 (d-k+1)^2 12 N \zeta \mu^{-k}. \tag{115}$$

*Proof.*

$$|f^*(x) - \hat{f}(x)| = \left| \sum_{l,m} a_{lm}^*(\sigma_{lm}(x) - \hat{\sigma}_{lm}(x)) \right| \tag{116}$$

$$\le d(d-k+1)\|a^*\|_\infty \sup_{lm} |\sigma_{lm}(x) - \hat{\sigma}_{lm}(x)| \tag{117}$$

$$\le (d+1)^2 (d-k+1)^2 12 N \zeta \mu^{-k}. \tag{118}$$

Thus,

$$(1 - \hat{f}(x)f^*(x))_+ \le |1 - \hat{f}(x)f^*(x)| \tag{119}$$

$$= |f^{*^2}(x) - \hat{f}(x)f^*(x)| \tag{120}$$

$$= |f^*(x)| \cdot |f^*(x) - \hat{f}(x)| \le (d+1)^2 (d-k+1)^2 12 N \zeta \mu^{-k}, \tag{121}$$

which implies the result. $\qquad\square$

## B.3    Second Step: Convergence

To conclude, we use one more time Theorem 6. Let $\mathcal{L}(a) := \mathbb{E}_{x \sim \mathcal{D}} \left[ L((a, b^1, w^1), \chi_{[k]}, x) \right]$. Then, $\mathcal{L}$ is convex in $a$ and for all $t \in [T]$,

$$\alpha^t = -\frac{1}{B} \sum_{s=1}^{B} \nabla_{a^t} L((a^t, b^1, w^1), \chi_{[k]}, x) + Z_{a^t}. \tag{122}$$

Thus, recalling $\sigma = \mathrm{Ramp}$, we have $\|\alpha^t\|_\infty \le 1 + \tau$ and $\|\alpha^t\|_2 \le \sqrt{N}(1 + \tau)$. Let $a^*$ be as in Lemma 14. Clearly, $\|a^*\|_2 \le 4(d-k+1)^{3/2}(d+1)^{1/2}$. Moreover, $a^1 = 0$. Thus, we can apply Theorem 6 with $\mathcal{B} = 4(d-k+1)^{3/2}(d+1)^{1/2}$ and $\xi = \sqrt{N}(1+\tau)$, and obtain that if:

- $T = \frac{64}{\epsilon^2}(d-k+1)^3(d+1)N(1+\tau)^2$;
- $\eta_t = \frac{\epsilon}{2N(1+\tau)^2}$ for $t \in \{1, \dots, T\}$;

then, with probability $1 - 3\delta$ over the initialization

$$\mathbb{E}_{x \sim \mathcal{D}} \left[ \min_{t \in [T]} L\left(\theta^t, \chi_{[k]}, x\right) \right] \le \frac{\epsilon}{2} + \frac{\epsilon}{2} = \epsilon. \tag{123}$$

## C    Proof of Theorem 5

Let us recall the definition of Cross-Predictability (CP) from [AS20].

**Definition 3** (Cross-Predictability (CP), [AS20]). *Let $P_{\mathcal{F}}$ be a distribution over functions from $\{\pm 1\}^d$ to $\{\pm 1\}$ and $P_{\mathcal{X}}$ a distribution over $\{\pm 1\}^d$. Then,*

$$\mathrm{CP}(P_{\mathcal{F}}, P_{\mathcal{X}}) = \mathbb{E}_{F, F' \sim P_{\mathcal{F}}}[\mathbb{E}_{x \sim P_{\mathcal{X}}}[F(x)F'(x)]^2]. \tag{124}$$

We invoke Theorem 3 from [AS20], which we restate here.

**Theorem 8** ([AS20]). *Let $\mathcal{P}_{\mathcal{F}}$ be a distribution on the class of parities on $d$ bits, and let $\mathcal{P}_{\mathcal{X}}$ be a distribution over $\{\pm 1\}^d$. Assume that $\mathbb{P}_{F \sim \mathcal{P}_{\mathcal{F}}, x \sim \mathcal{P}_{\mathcal{X}}}(F(x) = 1) = \frac{1}{2} + o_d(1)$. Consider any neural network $\mathrm{NN}(x; \theta)$ with $P$ edges and any initialization. Assume that a function $f$ is chosen from $\mathcal{P}_{\mathcal{F}}$ and then $T$ steps of noisy-SGD (Def. 1) with learning rate $\gamma$, gradient range $A$, batch size $B$ and noise level $\tau$ are run on the network using inputs sampled from $\mathcal{P}_{\mathcal{X}}$ and labels given by $f$. Then, in expectation over the initial choice of $f$, the training noise, and a fresh sample $x \sim \mathcal{P}_{\mathcal{X}}$, the trained network satisfies:*

$$\mathbb{P}(\mathrm{sgn}(\mathrm{NN}(x; \theta^T)) = f(x)) \leq \frac{1}{2} + \frac{TPA}{\tau} \cdot \left( \mathrm{CP}(\mathcal{P}_{\mathcal{F}}, \mathcal{P}_{\mathcal{X}}) + \frac{1}{B} \right)^{1/2} \tag{125}$$

In Theorem 8 we imported Theorem 3 from [AS20] with 1) the tighter bound for the specific case of parities, which gives a term $(\mathrm{CP} + 1/B)^{1/2}$ instead of $(\mathrm{CP} + 1/B)^{1/4}$ (see Theorem 4 and remarks in [AS20]); 2) the bound to the junk flow term $\mathrm{JF}_T$ for noisy-SGD with uniform noise, instead of Gaussian noise, which gives a term $P$ instead of $\sqrt{P}$ (see comments in [AS23]).

We thus compute the cross-predictability between $\mathcal{P}_{\mathcal{X}} = \mathcal{D}$ and $\mathcal{P}_{\mathcal{F}}$ being the uniform distribution on the set of parities of degree $k$.

**Lemma 16.** *If $\mu = \theta(1)$,*

$$\mathrm{CP}(\mathrm{Unif}\{k\text{-parities}\}, \mathcal{D}) \leq \binom{d}{k}^{-1} + C'_k \cdot \rho^2 \cdot \mu^{4k}, \tag{126}$$

*where $C'_k$ is a constant that depends only on $k$.*

*Proof.*

$$\mathrm{CP}(\mathrm{Unif}\{k\text{-parities}\}, \mathcal{D}) = \mathbb{E}_{S,S'} \mathbb{E}_{x \sim \mathcal{D}} \left[ \chi_S(x) \chi_{S'}(x) \right]^2 \tag{127}$$

$$= \binom{d}{k}^{-1} + \mathbb{E}_{S,S':S \neq S'} \mathbb{E}_{x \sim \mathcal{D}} \left[ \chi_S(x) \chi_{S'}(x) \right]^2 \tag{128}$$

$$= \binom{d}{k}^{-1} + \mathbb{E}_{S,S':S \neq S'} \rho^2 \mathbb{E}_{x \sim \mathcal{D}_\mu} \left[ \chi_S(x) \chi_{S'}(x) \right]^2 \tag{129}$$

$$\overset{(a)}{\leq} \binom{d}{k}^{-1} + \rho^2 \cdot \sum_{l=0}^{k-1} d^{-l} \mu^{4(k-l)} (C_k + O(d^{-1})) \tag{130}$$

$$= \binom{d}{k}^{-1} + \rho^2 \cdot \mu^{4k} \cdot \sum_{l=0}^{k-1} \left( \frac{1}{d\mu^4} \right)^l (C_k + O(d^{-1})) \tag{131}$$

$$\sim \binom{d}{k}^{-1} + C_k \cdot \rho^2 \cdot \frac{d^{-k} - \mu^{4k}}{(d\mu^4)^{-1} - 1}, \tag{132}$$

where in $(a)$ we used that for $S$ and $S'$ two randomly sampled sets of $k$ coordinates, we have:

$$\mathbb{P}(|S \cap S'| = l) = \frac{\binom{k}{l} \binom{d-k}{k-l}}{\binom{d}{k}} \tag{133}$$

$$= d^{-l} \left( \frac{k \binom{k}{l} \Gamma(k)}{\Gamma(1+k-l)} + O(1/d) \right) \tag{134}$$

$$\leq d^{-l} (C_k + O(1/d)), \tag{135}$$

where $C_k$ is such that $C_k \geq \frac{k \binom{k}{l} \Gamma(k)}{\Gamma(1+k-l)}$ for all $l \in [k-1]$.

If $\mu \gg d^{-1/4}$, then

$$\mathrm{CP}(\mathrm{Unif}\{k\text{-parities}\}, \mathcal{D}) \leq \binom{d}{k}^{-1} + C'_k \cdot \rho^2 \cdot \mu^{4k}, \tag{136}$$

where $C'_k$ depends only on $k$. $\qquad\square$

Finally, we need to discuss that Theorem 5 applies to any fully connected network with weights initialized from a distribution that is invariant to permutation of the input neurons. Let $\chi_S(x) = \prod_{j \in S} x_j$, with $|S| = k$, and let $\pi$ be a random permutation of the input neurons. Theorem 8 and Lemma 16 imply that after $T$ steps of training with noisy-SGD with the parameters $A, B$ and $\tau$, on data generated by $\mathcal{D}$ and $\chi_S$, the trained network is such that

$$\mathbb{E}_\pi \mathbb{P}(\mathrm{NN}(x; \theta^T) = (\chi_S \circ \pi)(x)) \leq \frac{1}{2} + \frac{TPA}{\tau} \left( \binom{d}{k}^{-1} + C_k' \rho^2 \mu^{4k} + \frac{1}{B} \right)^{1/2}. \qquad (137)$$

Recall that the network is fully connected and the weights outgoing from the input neurons are initialized from a distribution that is invariant to permutation of the input neurons. Consider the action induced by $\pi$ on the weights outgoing from the input neurons. By properties of SGD, it follows that each conditional probability on $\pi$ contributes equally to the left hand side of (137), thus the same bound holds also for the single function:

$$\mathbb{P}(\mathrm{NN}(x; \theta^T) = \chi_S(x)) \leq \frac{1}{2} + \frac{TPA}{\tau} \left( \binom{d}{k}^{-1} + C_k' \rho^2 \mu^{4k} + \frac{1}{B} \right)^{1/2}. \qquad (138)$$

## D  Proof of Proposition 1

Note that

$$1 - y^s \bar{y} - \hat{f}(x^s)(y^s - \bar{y}) = (y^s)^2 - y^s \bar{y} - \hat{f}(x^s)(y^s - \bar{y}) \qquad (139)$$
$$= (y^s - \hat{f}(x^s))(y^s - \bar{y}) \qquad (140)$$
$$= (1 - y^s \hat{f}(x^s))(1 - y^s \bar{y}) \qquad (141)$$

Since $|\bar{y}| < 1$, $\mathrm{sgn}(1 - y^s \bar{y}) = 1$, and $\mathrm{sgn}((1 - y^s \hat{f}(x^s))(1 - y^s \bar{y})) = \mathrm{sgn}(1 - y^s \hat{f}(x^s))$. Thus,

$$\epsilon > \frac{1}{m} \sum_{s \in [m]} \left( (1 - y^s \hat{f}(x^s))(1 - y^s \bar{y}) \right)_+ \qquad (142)$$
$$= \frac{1}{m} \sum_{s \in [m]} \left( (1 - y^s \hat{f}(x^s))(1 - y^s \bar{y}) \right)_+ \mathbb{1}(y^s \hat{f}(x^s) < 1) \qquad (143)$$
$$\geq \frac{1}{m} \sum_{s \in [m]} \left( (1 - y^s \hat{f}(x^s))(1 - y^s \bar{y}) \right)_+ \mathbb{1}(y^s \hat{f}(x^s) < 0) \qquad (144)$$
$$\geq \delta \frac{1}{m} \sum_{s \in [m]} \mathbb{1}(y^s \hat{f}(x^s) < 0). \qquad (145)$$

## E  Experiment Details and Additional Experiments

In this section, we provide more details on the implementation of the experiments. Moreover, we present further experiments on the comparison of the curriculum strategy and standard training.

### E.1  Experiment Details

#### E.1.1  Architectures and Loss Functions

**Architectures.**  We used an MLP model trained by SGD under the $\ell_2$ loss for the results presented in the main part. In this appendix, we further present experimental results using other models, particularly, mean-field [MMN18] and Transformer [VSP+17]. Below, we describe each in detail.

- **MLP.** The MLP model is a fully-connected architecture consisting of 4 hidden layers of sizes 512, 1024, 512, and 64. The ReLU activation function has been used for each of the layers (except the last one). Moreover, we have used PyTorch's default initialization which initializes weights of each layer with $U(\frac{-1}{\sqrt{\dim_{in}}}, \frac{1}{\sqrt{\dim_{in}}})$ where $\dim_{in}$ is the input dimension of the corresponding layer.

- **Mean-field.** We also use a two-layer neural network with mean-field [MMN18] parametrization. Particularly, we define $f_{\mathrm{MF}}(x) = \frac{1}{N}\sum_{i=1}^{N} a_i \sigma(\langle w_i, x\rangle + b_i)$ where $N = 2^{16}$ and $\sigma = \mathrm{ReLU}$ are the number of neurons and the activation function respectively. We also initialize the weights following $a_i \sim U(-1,1)$, $w_i \sim U(\frac{-1}{\sqrt{d}}, \frac{-1}{\sqrt{d}})^{\otimes d}$, and $b_i \sim U(\frac{-1}{\sqrt{d}}, \frac{-1}{\sqrt{d}})$, where $d$ is the dimension of the input. Note that with this parametrization, one should employ large learning rates (e.g., 1000) assuming SGD is used as the optimizer.

- **Transformer.** We use the standard encoder part of Transformer architecture [VSP$^+$17] which is commonly used in language modeling ([RSR$^+$19]) and vision applications (particularly, Vision Transformers [DBK$^+$20]). More specifically, we first embed $\pm 1$ into a 256-dimensional vector using a shared embedding layer. Then, the embedded input is processed through 6 transformer layers. In each transformer layer, the size of the MLP hidden layer is also 256, and 6 attention heads are used. Finally, a linear layer is utilized for computing the output. We also use learnable positional embedding.

**Loss Functions.** We have mainly used the $\ell_2$ loss for our experiments. Nonetheless, we additionally present results on the hinge and covariance loss which are used in our theoretical results. We briefly review each of these losses below.

- **$\ell_2$ loss.** We use the squared loss $(\hat{y} - y)^2$ for each sample where $\hat{y}$ and $y$ are the predicted and the true values respectively.

- **Hinge loss.** We use the hinge loss defined as $\max(0, 1 - \hat{y}y)$ for each sample where the true label $y \in \{\pm 1\}$ and the predicted value $\hat{y} \in \mathbb{R}$.

- **Covariance loss.** We also use the covariance loss as defined in Definition 2, i.e., $\max(0, (y - \bar{y})(y - \hat{y})) = \max(0, 1 - y\bar{y} - \hat{y}(y - \bar{y}))$, where $y \in \{\pm 1\}$, $\bar{y} \in (-1, 1)$ and $\hat{y} \in \mathbb{R}$ present the true label, the average of the true label (on the training distribution), and the predicted value.

### E.1.2 Procedure

We have implemented the experiments using the PyTorch framework [PGM$^+$19]. The experiments were executed (in parallel) on NVIDIA A100, RTX3090, and RTX4090 GPUs and consumed approximately 200 GPU hours (excluding the selection of hyperparameters). Note that each experiment has been repeated using 10 different random seeds and the results are reported with $95\%$ confidence intervals.

**Curriculum Strategy / Standard Training.** Here we explain the general setting of training. When using the curriculum strategy, the model is first trained on the sparse samples until the training loss reaches below $10^{-2}$. Then, in the second phase of the curriculum, the training is continued on the mixed distribution until the training loss drops below $10^{-3}$. On the other hand, during the standard training, the training is done on the mixed distribution, again, until the training loss becomes less than $10^{-3}$.

**Number of Samples/Number of Iterations.** We generally focused on two components to compare the curriculum and standard training: sample complexity and the number of iterations needed for convergence. To compare the sample complexity, for a given number of training samples, we first generate a training set according to $\rho$ and $\mu$. In the case of using the curriculum, the sparse samples are then selected by checking if their Hamming weight satisfies $H(x) < d(\frac{1}{2} - \frac{\mu}{4})$. Afterward, training is done as described earlier with or without the curriculum. Finally, the test accuracy (or the test loss) is compared. We are also interested in the number of steps needed for the convergence of models whether the curriculum is used or not. To study the number of optimization steps independently of the number of samples, we consider an online setting in which each mini-batch is freshly sampled and there is no fixed training set. Note that in this case, when in the curriculum's first phase, we directly sample the sparse distributions from $\mathcal{D}_\mu$. Similarly, the model is trained until the training loss drops below $10^{-3}$, and then the number of iterations is compared. The only exception is $f_{\mathrm{middle}}$ in Figure 3, where we were mainly interested in the weak learning of the target function. In that case, the training is stopped when the training loss reaches below $0.26$.

**Hyperparameter Tuning.** Note that the experiments are aimed at comparing the standard and curriculum-based training in a fair and similar context. As a result, we did not perform extensive hyperparameter tuning for our experiments. (Also note that the task and/or the training distribution varies from one experiment to another.) Nonetheless, we tried different batch sizes and learning rates to ensure the robustness of our findings. Generally, we did not observe any significant change in the patterns and potential gains of the curriculum method (of course changes like using a smaller learning rate would make both curriculum and standard training slower). Consequently, we used a moderate batch size of 64 for all of our experiments (other than Transformers) and we selected the learning rate based on the speed and the stability of the convergence. Particularly, we used a learning rate of $0.003$ when optimizing the MLP model with SGD under the $\ell_2$ loss (which covers all experiments presented in the main part). For the mean-field model, we use a learning rate of 1000. Also for the experiments of Transformers, we used the Adam [KB14] optimizer with batch size 512 and learning rate 0.0001.

## E.2 Additional Experiments

First, we complete Figure 2 by presenting the plot that shows the gain of the curriculum method in the number of iterations needed for convergence for different parity degrees in Figure 4. Similar to the previous experiments, we consider the parity embedded in dimension $d = 100$. As expected, it is shown that parities of higher degree require more iterations to be learned in general. However, this increase in iterations is mild when the curriculum is employed, which leads to an increasing gap between the curriculum and standard training as the degree of parity is increased.

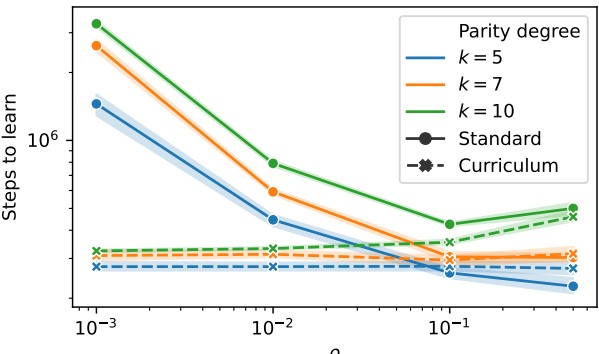

Figure 4: Dependency of the gains of the curriculum strategy in the number of training steps for different degrees of the parity function. The gain of the curriculum method is potentially increased if parities of higher degrees are used.

Next, we move to the mean-field model. Similar to the other experiments, we evaluate the benefit of the curriculum strategy for both the sample complexity and the number of steps needed for convergence. We empirically observed that the mean-field model is able to learn parity functions more easily (with fewer samples and iterations) than the MLP model used in the main text. We thus use parity of degree $k = 7$ in dimension 100 and $\mu = 0.98$ to demonstrate the gains of the curriculum strategy more clearly. Particularly, we report the accuracy of the mean-field model trained under the $\ell_2$ loss for $\rho = 0.01$ and different training set sizes in Figure 5 (left). Likewise, the number of iterations needed for convergence for different values of $\rho$ is presented in 5 (right). As expected, the gap between curriculum strategy and standard training in both the sample complexity and number of optimization steps are similarly exhibited if a mean-field model is used.

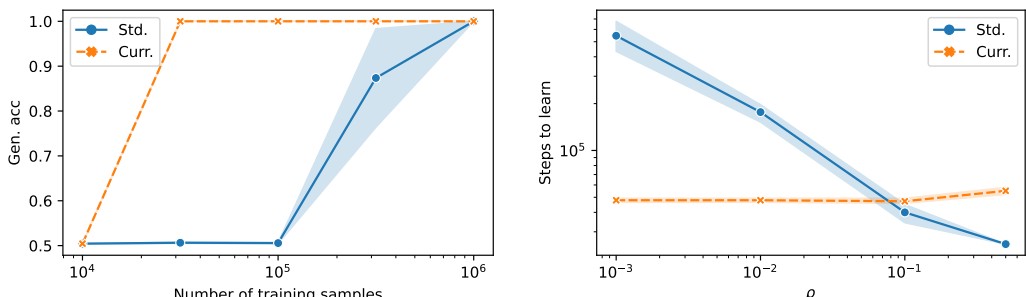

Figure 5: Comparison of curriculum and standard training for the mean-field model trained under the $\ell_2$ loss; (left) accuracy of the model trained on training sets with varying sizes, (right) the number of iterations needed for convergence of the model. The general picture of the potential benefits of the curriculum method remains unchanged.

Moreover, we study other loss functions studied in this paper. To this end, we keep the same setting as above, considering the mean-field model, parity of 7 bits, and $d = 100$, $\mu = 0.98$, and $\rho = 0.01$ (unless varying). We report the results of the hinge and covariance loss in Figure 6 and Figure 7 respectively. It can be seen that the patterns are generally similar to the case that $\ell_2$ loss was being used (Figure 5). In sum, the curriculum method makes the learning of the parity function possible with fewer samples and iterations for small enough values of $\rho$.

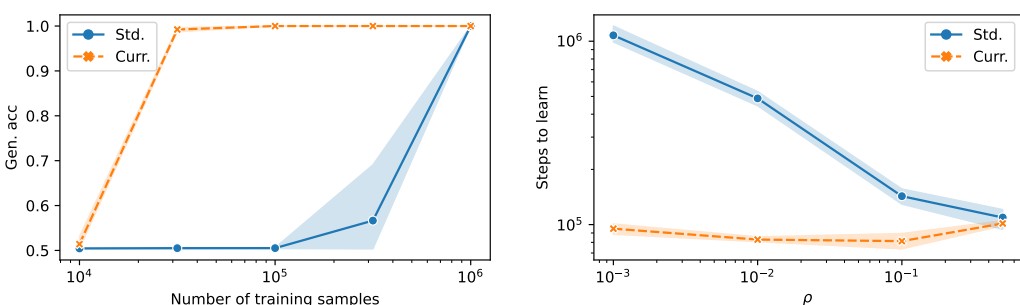

Figure 6: Potential benefits of the curriculum strategy for the mean-field model trained with the hinge loss; (left) accuracy for different training set sizes, (right) the number of iterations (with fresh samples) needed for the convergence of the model. Similar to the case of the $\ell_2$ loss, the curriculum is beneficial for small enough values of $\rho$.

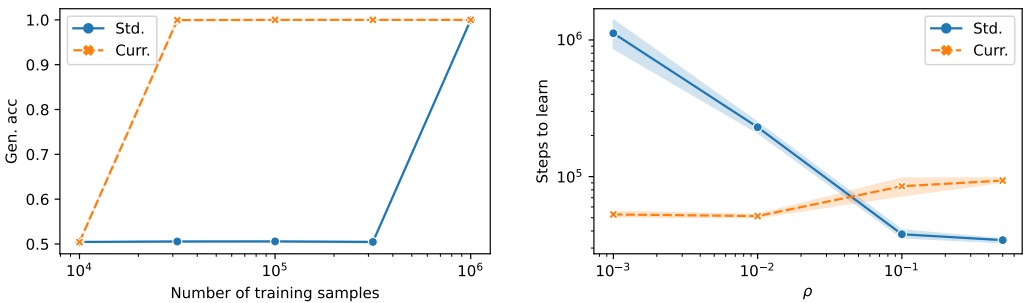

Figure 7: Comparing curriculum strategy and standard training for the mean-field model trained with the covariance loss; (left) accuracy for different training set sizes, (right) the number of optimization steps needed for learning the parity function. Similar to experiments where $\ell_2$ or hinge loss are used, curriculum potentially reduces the number of iterations and also the number of samples needed for learning.

Finally, we evaluate the curriculum strategy for the Transformer model trained under the $\ell_2$ loss. To make the optimization of the model faster, we use Adam [KB14] optimizer with batch size 512 for this experiment. We also stop the training when the training loss reaches below $10^{-2}$ instead of $10^{-3}$. Similar to the above-mentioned setting, we set $\mu = 0.98$, $\rho = 0.01$ (unless varying), $d = 100$, and we consider learning the parity of degree $k = 7$. The validation accuracy of the Transformer for different training set sizes is presented in Figure 8 (left). It can be seen that there is a significant, albeit weak, advantage in terms of sample complexity when the curriculum is used. Similarly, the number of iterations with fresh samples needed for convergence of the model is shown in Figure 8 (right). The gain in the number of steps is also weaker than the other models, nonetheless, it is significant (especially for small values of $\rho$ such as 0.001).

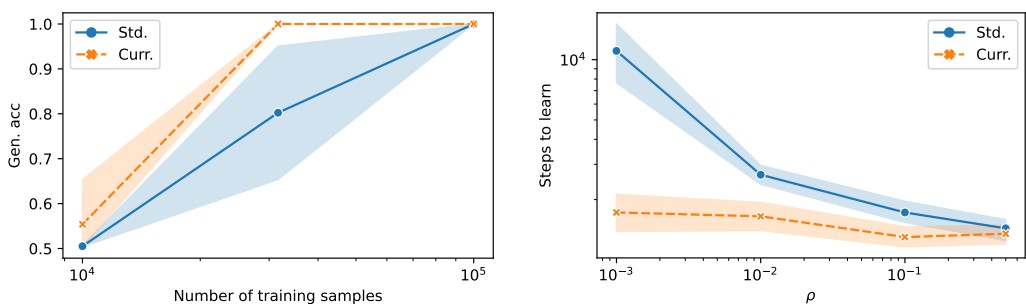

Figure 8: Evaluating the curriculum for the Transformer model in terms of sample complexity (left) and the number of iterations (right). It can be seen that the curriculum can still be significantly beneficial, although the difference between the curriculum and standard training is weaker in comparison to the MLP and mean-field model.

