# OpenReview forum: "Provable Advantage of Curriculum Learning on Parity Targets with Mixed Inputs"
_NeurIPS.cc/2023/Conference — NeurIPS 2023 poster_

### Official Review · Reviewer_3jFh · 2023-07-03

**Soundness:** 3 good
**Presentation:** 3 good
**Contribution:** 3 good
**Rating:** 6
**Confidence:** 4

**Summary:**

This paper presents a theoretical analysis of curriculum learning, in which a neural network first presented with “easier” examples is able to more efficiently learn a target over more “complex” examples. The authors specifically consider the problem of learning $k$-sparse parities over a distribution which is a mixture of the uniform distribution over the hypercube and a biased distribution over the hypercube (the easy examples). The main positive results study a two-layer neural network trained via a layer-wise curriculum version of SGD (Algorithm 1), where first the first layer weights are trained on samples from the biased distribution, and next the second layer weights are trained on samples from the mixture distribution. Theorems 3 shows that this algorithm learns $k$-sparse parities with respect to the mixture distribution in $\tilde O(d)$ steps of SGD. In contrast, the paper presents a lower bound showing that noisy SGD on the mixture distribution alone requires $\Omega(d^{1+\delta})$ timesteps to learn a $k$-sparse parity up to nontrivial error. The proof for the upper bound relies on the martingale plus drift argument to show that the $k$ relevant coordinates have been learned after the first stage, while the lower bound uses similar techniques to proving SQ lower bounds.

**Strengths:**

- The paper is well written.
- I skimmed the proofs in the appendix and they appear to be sound. The experiments section demonstrates that the proposed curriculum strategy is effective in settings beyond those considered in the theory.
- Curriculum learning is a common procedure in practice, and yet there is limited theoretical understanding of its behavior. The sparse parity problem presented here is an interesting setting to study curriculum learning, with a clear measure of “easier” samples. The results here adequately show that a neural network can obtain an efficiency improvement from first training on these easier samples, which I find to be a very nice result.

**Weaknesses:**

- The novelty of the paper, in comparison to the prior work [CM23], is a bit difficult to discern. It appears that the main difference is that the current paper uses SGD for multiple steps in the first stage and uses the “drift plus martingale” technique for the proof, while [CM23] considers a single large step of GD with large batch size in the first stage and shows that is sufficient for learning the support. However, Theorem 3 here seems very similar to Theorem 3.4 in [CM23]. Also, Theorem 4 still uses the one-step algorithm. Could you please comment further on the novelty of the current paper in comparison to the prior work?
- A minor weakness is that the algorithm is a bit restrictive and has some nonstandard modifications. In particular, during the first stage the biases are chosen to be very large ($\Theta(d)$), and then resampled to different deterministic values before the second stage. Next, there is an $\ell_\infty$ projection on the weights during each step of stage 1. Could you please comment on the necessity of these modifications? I think the paper could benefit from additional discussion of these modifications.

**Questions:**

- It appears that in Theorem 3, the algorithm receives far more samples from the biased distribution than the uniform distribution. In particular, during the first stage the algorithm receives $\tilde O(d)$ biased samples, while in the second stage receives $\tilde O(d)$ uniform samples. In contrast, when running SGD on the entire mixture distribution (like for the lower bounds in Theorem 5), the algorithm receives only $\rho$-times as many biased samples as uniform samples. In the regimes where there is a separation, $\rho$ is very small, in which case curriculum SGD is receiving far more samples from $D_\mu$. It thus seems like the success of curriculum SGD is not necessarily due to viewing the “easy” examples first, but rather from having access to far more “easy” examples than in regular SGD. Is my understanding here correct, and can you please comment on this further?

I am open to increasing my score after further clarifications on the above point and my concerns in the weaknesses section.

- Minor typos: In Algorithm 1, should line 3 be $\frac12 - \frac{\mu}{4}$ instead?



**Limitations:**

Limitations are adequately addressed.

---

> ### Author Rebuttal · Authors · 2023-08-09
>
> Please see the global response for the differences with the previous work [CM23].
>
> - Q1: A minor weakness is that the algorithm is a bit restrictive and has some nonstandard modifications. In particular, during the first stage the biases are chosen to be very large Theta(d), and then resampled to different deterministic values before the second stage. Next, there is an $l_\infty$ projection on the weights during each step of stage 1. Could you please comment on the necessity of these modifications? I think the paper could benefit from additional discussion of these modifications.
>
> A1: Biases: We consider fixed bias parameters for simplicity of the theoretical analysis. Regarding the specific values that we chose, we initialized them to large constants in order to have simple expressions for the gradients of the network during the first part of training (eq. (25)). After the first part of training, we update the biases so that they are well spread in the interval $[-\Delta k, \Delta k]$ and the target parity belongs to the linear span of the hidden units at time $T_1$.
>
> $l_\infty$ projection: We added an $l_\infty$ projection to prevent weights from diverging. This projection is needed because the training process doesn't occur simultaneously for both layers. As a result, the training doesn't stop once the network fits the data.
>
> We will add a discussion of the above in the paper.
>
> - Q2: It appears that in Theorem 3, the algorithm receives far more samples from the biased distribution than the uniform distribution. In particular, during the first stage the algorithm receives Theta(d) biased samples, while in the second stage receives Theta(d)  uniform samples. In contrast, when running SGD on the entire mixture distribution the algorithm receives only $\rho$-times as many biased samples as uniform samples. It thus seems like the success of curriculum SGD is not necessarily due to viewing the “easy” examples first, but rather from having access to far more “easy” examples than in regular SGD.
>
> A2: It is true that our current theoretical results cannot distinguish between having more easy samples or less. Indeed although we have proved an upper bound on the number of samples needed, we do not have any lower bound on the sample complexity. We have tried to explain this in Remark 4. Nonetheless, in the sample complexity experiments (Figure 1)  we are using the exact same data and we only change the ordering for curriculum and standard training.
>
> - Q3: Minor typos: In Algorithm 1, should line 3 be $1/2-\mu/4$ instead?
>
> A3: Yes, thank you for pointing out the typo

---

> > ### Comment · Reviewer_3jFh · 2023-08-11
> > **Response to Authors**
> >
> > Thank you to the authors for their response. The contribution of the current submission in comparison to [CM23] is now more clear to me, and so I have increased my score accordingly. I think it is still interesting to understand whether the benefit of curriculum learning shown here is really due to the order of the examples, or rather just having access to more easy samples.

---

> > > ### Author Response · Authors · 2023-08-20
> > >
> > > Thank you for considering the rebuttal and increasing your score. We agree that it would be interesting to prove the benefits of the proposed curriculum strategy regarding sample complexity as well. (We emphasize that the sample complexity experiments already show this empirically.)

---

### Official Review · Reviewer_8Apd · 2023-07-06

**Soundness:** 3 good
**Presentation:** 2 fair
**Contribution:** 3 good
**Rating:** 6
**Confidence:** 3

**Summary:**

The authors present a separation result between curriculum learning and standard learning in the number of noisy-(S)GD training steps for a 2-layer ReLU network for case of learning noiseless k-parity. While the precise training algorithm deviates from the common deep learning setup, experiments verify that learning k-parity can benefit from learning sparse inputs before harder dense inputs - mostly in terms of an improved sample complexity.


**Strengths:**

- Theoretical evidence for benefits of curriculum learning.
- Additional positive results for the hinge loss.
- Bound on sample complexity for the positive case.
- The negative result allows for a comparison with the positive one in some settings.
- The experimental results suggest a strong difference between curriculum learning and standard learning (yet, not exactly in the theoretical setup so far) in the context of learning a parity function.

**Weaknesses:**

- The analysis is limited to a single and not very general learning problem (learning k-parity).

- The analysis makes assumptions that are quite different from practical deep learning setups:
1) Non-standard neural network initialization.
2) Non-standard loss (covariance loss) that penalizes errors on samples with true negative labels higher than errors on truly positive samples. The hinge loss results rely on an uncommon activation function.
3) The hinge loss results are relatively weak as they just state the existence of an initialization etc. that leads to successful training. Essentially, this is not stronger than an expressiveness results, because the initialization could just correspond to the optimal (potentially learned) parameters.
4) No noise in the data.
5) Curriculum learning and dense learning use learning algorithms that are different also in other aspects than just the order of samples. For instance, the batch size is required to be quite large in case of dense learning. Most importantly, the theorem for curriculum learning assumes a much larger proportion of easy examples $\rho$ than the standard learning. This is important because $\rho$ essentially controls how easy the learning problem is. In the experiments, for large enough $\rho$ dense learning seems to perform better than curriculum learning. (Btw, this observation is not explained by the theory.)
6) The learning algorithm is not standard SGD or GD but learns the layers separately.
7) The bias parameters are not trained (and it is left unclear in the main paper how they are initialized).

- Mismatch between experimental validation and theory:
1) different loss (l2 in experiments, mostly covariance loss in theory)
2) k = 5 in most experiments, while theory assumes k >=6 for separation result.
3) MLP with four layers instead of 2.
Experiments should also validate the theory and not only study fairly unrelated cases where we can make similar observations. The direct validation would be important to understand how relevant the constants in the theoretical results are, for instance.

- The theorem statements are hard to understand because they rely on notation that has not been introduced in the main text. (Examples: $\mu^{k-1}$ or $\Delta$ or P.)

- The proof techniques do not appear to be very novel but similar to previous work.

- Missing information: How is the generalisation accuracy computed? I assume samples were drawn from the mixed data distribution, which depends on $k$ and $\rho$? How many samples were used in the experiments?
This is important because it also influences how hard the learning problem is.

- The theoretical and experimental results study the performance with respect to a test distribution that depends on the amount of easy versus hard examples.
This is suboptimal because it can therefore make no claims about the question what kind of data would benefit learning the actual parity function (on all possible inputs).
In contrast, the central question underlying curriculum learning is what kind of data (and which order) is best for learning a task.


**Points of minor critique:**

- Literature discussion:
Given the literature discussion, the study of the specific problem of learning parities in the context of curriculum learning does not seem to be in its infancy, as claimed.
Furthermore, [SMS22] does not really make the difficulty level criterion label dependent but dependent on the (irrelevant features). In some sense, the same holds also true for the submitted work, as the definition depends on the negative coordinates that also inform the label.
(Yet, this does not change the fact that [SMS22] and the presented work are sufficiently different.)

- More elements of the notation could be explained. For instance, Rad could be introduced as Rademacher distribution; Unit as uniform distribution; neural networks could be defined formally earlier, []_A could be defined mathematically, etc. The bias scale was not defined.
How are the biases initialized?
The size P of a neural network has not been defined precisely. Does it refer to the number of hidden neurons or number of parameters?
Weak learning has not been defined/explained.

- The set of sparse inputs does not have size $\rho$ on average according to the definition of $X_1$ on page 3 (different from how it is frequently discussed). Also $D_u$ contributes to $X_1$, while $D_{\mu}$ does not always. Later, $d$ is assumed to be large enough that the statement is approximately correct, but that comes a bit late (and is not very precise).

- Only even values of $\mu$ have been studied in the experiments. Uneven averages should also be considered.

- Not all curves in Figure 2 (middle and right) are visible.


**Questions:**

- To which degree does the performance actually depend on being exposed to the hard samples?
Curriculum learning is based on the hypothesis that it is not only beneficial to learn from easy examples early but that learning from hard samples later also boosts the performance.
I am aware the the negative result needs $\rho$ to be small. Yet, I suspect that learning only on "easy" examples would already be sufficient to learn a parity function because the easy examples carry all relevant information about the parity function.
To analyze this question in more detail, it would be good to 1) make the generalisation accuracy independent of the mixed distribution and test whether the correct target function has been learned; 2) report how the performance changes during training.

- Figure 2 (middle) implies a non-monotonous dependence of the accuracy in $\mu$. It would be good to study this in more detail and show a figure with $\mu$ on the x-axis. What could be an explanation? Does it matter whether $d\mu$ is even or uneven?

- To which degree are the experimental results dependent on hyperparameter tuning? Where the standard learning and curriculum learning be tuned independently?



**Limitations:**

The limitations have been pointed out under weaknesses and were mostly discussed by the authors. I do not foresee a major or immediate societal impact of this work.

---

> ### Author Rebuttal · Authors · 2023-08-09
>
> - Q1. What about training only on easy samples?
>
> A1. Note that neural networks do not know that the target function is a parity function a priori, thus, they cannot recover the parity function only from the sparse data. (We agree that if the learner knew that the target is k-parity it would have been able to recover it only from the sparse samples, as identifying the support would have been sufficient.) Indeed, we observed that mismatch between the training distribution and test distribution would result in worse generalization and that training neural networks on sparse samples would cause the neural network to learn other lower-degree solutions rather than the actual parity function.
> - Q2. Non-monotonous dependence of the accuracy in $\mu$.
>
> A2. We note that this non-monotonous dependency is indeed reflected in our theoretical results as well. Particularly, the results in Theorem 3 depend on the quantity $L_\mu = \mu^{k-1} - \mu^{k+1}$, which is maximized near $\mu = 1$  but does not increase monotonically with $\mu$.
> Due to the challenges of defining sample complexity properly, we rather not to plot the sample complexity with having $\mu$ on the x-axis. Please also check Figure 4 which also shows the dependency on $\mu$.
> Finally, we remark that $d\mu$ is a real-valued and continuous quantity; we thus do not expect any different behavior regarding evenness. We will add more context regarding the dependency of the sample complexity on $\mu$.
> - Q3. Hyperparameter tuning.
>
> A3. We tried different values for hyperparameters (mainly batch size and learning rate) to ensure the robustness of our results and we did not observe any significant changes in the potential gains of the curriculum method. For the experiments presented in the paper, we selected hyperparameters based on the speed and stability of the convergence of the standard training. (We tried to use values that are common in practice as well, e.g., batch size of 64). To ensure fairness, we used the values that we picked for standard training for the curriculum method as well.  We have explained our hyperparameter tuning in Appendix E.1.2.
> - Q4. Limitation to parity targets.
>
> A4. Indeed, the focus of this paper is on parity targets. However, we experimentally investigated the proposed curriculum method for Boolean functions other than parity targets (see the last part of Section 5). Our experiments suggest that the proposed curriculum method can still be beneficial for learning or weak learning some Boolean functions, but a deeper understanding of those is left to future work. However, we have also shown that there are functions for which the curriculum is not helpful.
>
> - Q5: Different assumptions from practical deep learning setups.
>
> A5: For the positive result, we have made some simplifying assumptions on the training settings so that the theoretical analysis becomes feasible. Note that some of these assumptions, such as layer-wise training, are quite common in the theory of deep learning. On the other hand, our negative result holds for all fully connected architectures of any depth, and with any activation and initialization that is invariant to permutations of the input neurons. Also, in our experiments we tried to use common deep learning settings.
> - Q6:  Hinge loss result.
>
> A6:  In Theorem 4, we prove that there exists an initialization such that, under appropriate hypothesis, SGD can successfully learn *any* $k$-parity. Thus, this is not just expressiveness, as one does not know which $k$-parity is the correct target and our initialization is agnostic of the target function. We will clarify the statement of the theorem to avoid this confusion and we thank the reviewer for pointing this issue out.
>
> - Q7. Curriculum learning and dense learning use learning algorithms that are different also in other aspects than just the order of samples.
>
> A7: We addressed this question in the general rebuttal.
> - Q8: Mismatch between experimental validation and theory.
>
> A8: We cover more general and common settings in our experiments comparing to the theoretical part. (As pointed out by the reviewer, some of our assumptions for the theoretical side are not common in practice, for this very reason we focused on the common training settings for the practice). Nonetheless, we emphasize that this is an extension of our theoretical results and not a mismatch. Additionally, we have also covered the setting of our theoretical results in the appendix. Particularly, in Figures 5,6,7 we have covered 2 layer neural network (with mean-field parametrization), hinge loss and covariance loss. We have also used parities beyond degree 5 in Figures 2 and 4.
> - Q9. How is the generalisation accuracy computed? Number of training samples?
>
> A9. The generalization accuracy is computed based on $D_{mix}$. In Q1 we explain the need for using the same distribution for training and test. The number of training samples is shown on the x-axis for experiments regarding sample complexity. For other experiments that work with fresh samples and number of iterations the number of training samples would be given by batch size (almost always 64) * number of iterations. However, we once again emphasize that samples are fresh and only seen once.
> - Q10. What kind of data is most useful?
>
> A10. Note that for each single experiment (or theoretical result) the training (=test) distribution is shared between curriculum and standard training to make the comparisons fair. In our setting, we need sparse data to start learning the function (detecting the latent dimensions) and we also need to have matching train and test distributions so sparse samples alone are not enough (see Q1 as well). This indeed shows why such a curriculum is helpful.
> - Q11. Points of minor critique
>
> A11. We thank the reviewer for these points. We will revise the text to clarify the points raised. We will clarify the notation and how the biases of Thm 3 are initialized in the main.

---

> > ### Comment · Reviewer_8Apd · 2023-08-14
> > **Score update**
> >
> > I thank the authors for the clarifications and pointing me to additional experiments in the appendix. I have increased by score to a weak accept in response.

---

> > > ### Author Response · Authors · 2023-08-20
> > >
> > > Thank you for considering the rebuttal and increasing your rating.

---

### Official Review · Reviewer_Sb2P · 2023-07-07

**Soundness:** 3 good
**Presentation:** 3 good
**Contribution:** 3 good
**Rating:** 6
**Confidence:** 1

**Summary:**

This paper presents provable results showing the efficiency of curriculum learning for a specific problem setting with training data $(x,y)$, where $x\in(\pm1)^d$ is mixed distributed and $y=\Pi_{j\in\mathcal{S}} x_j\in(\pm1)$. What's more, the study utilizes a 2-layer fully connected network and the noisy-SGD training method

**Strengths:**

1. This paper clearly proves the advantages of curriculum learning over standard learning by analyzing a well-defined problem.
2. Numerous results are also provided and highlight the reduced sample and iteration requirements of curriculum learning.

**Weaknesses:**

Please see Questions.

**Questions:**

1. This paper introduces a highly specific data setting. How can this setting be applied to real-world applications or more general settings?
2. Is there a particular significance to the notion of "sparsity"? I am curious about the evidence if the model was trained on dense data instead of sparse data initially.
3. In Theorem 3, why is $T_1$ independent to the first layer model size $Nd$, while $T_2$ is much larger than the second layer size $N$? Additionally, is the result evaluated using training accuracy?
4. In my opinion, a more appropriate point of comparison for standard training would be: **randomly** sample an equivalent dataset size (as in curriculum learning) to train the first layer, followed by training the second layer using the entire dataset. Since the first layer contains significantly more parameters, standard training might introduce overfitting due to the layer-wise training method rather than curriculum learning.

**Limitations:**

Yes

---

> ### Author Rebuttal · Authors · 2023-08-09
>
> - Q1. This paper introduces a highly specific data setting. How can this setting be applied to real-world applications or more general settings?
>
> A1. In our opinion, the closest curriculum method in the real world is the use of input length for NLP and reasoning tasks. Note that $+1$ is the identity element in the Boolean setting and thus can also be viewed as a padding element. Consequently, the number of $-1$’s can be seen as the length of the Boolean inputs (see related literature). We also expect similar results to be found in the learning of functions with Gaussian inputs. However, the goal of this paper is to prove the benefit of curriculum learning for a problem (with the same task on the same data distribution) where theoretical analysis is possible, and that is the reason that we have focused on the parity functions where we have been able to prove a separation result.
>
> - Q2. Is there a particular significance to the notion of "sparsity"? What about training the model on the dense data initially?
>
> A2. On sparse data (i.e., inputs with few negative bits), parities on different supports are correlated. In particular, $\mathbb{E}_x[\chi_S(x)\chi_T(x)] = \rho \mu^{|S \Delta T|}$ assuming $S\neq T$ ($\Delta$ stands for symmetric set difference, i.e., union minus intersection). This allows SGD to easily identify the relevant coordinates, and a subsequent fit on the whole dataset allows to learn the correct function. We remark that the same holds if we defined sparse inputs as those with few $+1$ bits, and mostly $-1$ bits (having negative $\mu$ – note that the absolute value of the quantity above does not change). On dense data (i.e., inputs with roughly half negative bits), parities on different supports are not correlated ($\mathbb{E}[\chi_S(x)\chi_T(x)] \approx 0$), which makes identifying the support challenging for any progressive learner (including SGD). For this reason, training initially on dense inputs only does not help for identifying the support of the parity, thus we expect such curriculum not to be beneficial for learning parities.
>
> - Q3. In Theorem 3, why is $T_1$  independent to the first layer model size $Nd$, while $T_2$ is much larger than the second layer size? Additionally, is the result evaluated using training accuracy?
>
> A3. The model size, and more specifically the number of hidden units, has to guarantee that the network can represent any $k$-parity. During the first layer training the network identifies the set of relevant coordinate, and $T_1$ does not depend on the number of parameters (note that at each step of training all weights in the first layer move). For the training of the second layer, we used standard results on convergence of SGD on convex losses. $T_2$ depends on $N$, however it is larger that that since we took a small learning rate ($\gamma_2 \approx 1/ Nd$). The result is evaluated using the training loss (covariance loss), which in our case can be related to the training accuracy (see Proposition 1). Note that in the experiments (other than the ones in Figure 3), we directly report accuracy.
>
> - Q4. Using first layer then second layer training for both the curriculum and standard training; since the first layer contains more parameters, standard training might introduce overfitting due to the layer-wise training method rather than curriculum learning.
>
> A4. We remark that the benefits of the curriculum algorithm is not due to the layer-wise training. First note that in the experiments we train all the layers jointly for both the curriculum and standard training. For the theoretical results, the negative result for standard training (Theorem 5) is also valid whether the training is done layer-wise or jointly. We will clarify this issue in the paper as well.

---

> > ### Comment · Reviewer_Sb2P · 2023-08-15
> >
> > Thank you to the authors for their thoughtful response and clarification. After considering the other reviews, I have decided to maintain my original rating.

---

> > > ### Author Response · Authors · 2023-08-20
> > >
> > > Thank you for reading our rebuttal and acknowledging it.

---

### Official Review · Reviewer_RLGW · 2023-07-25

**Soundness:** 3 good
**Presentation:** 4 excellent
**Contribution:** 2 fair
**Rating:** 6
**Confidence:** 5

**Summary:**

This paper studies the impact of curriculum learning for the family of parity functions. Compared to previous work, in which their settings have been distanced from a realistic setting, e.g., considering a non-common activation function with a particular initialization, or a non bounded learning rate, this paper solves all of those issues, although still in special case of covariance loss. They have been successful to show how the order of samples given to the model will effect the sample complexity, both in theory and experiments. Besides, to the downsides of the standard SGD algorithm that samples are completely shuffled, they have provided quite tight upper bounds for the final accuracy of a general fully connected neural network with limited total size, when assuming the procedure is noisy to make it resemble SQ-learning setting and use similar proofs.

**Strengths:**

1- Authors' suggested setting has been closest to reality so far. Unlike previous work they use a bounded learning rate throughout the training process, whereas the former algorithm had proceeded by using only one step of sgd with a learning rate that can get arbitrarily large.

2- The combination of a lower bound for test accuracy when using curriculum learning (giving correlated samples at first so that it approximately learns the support) and an upper bound in case of standard random batch sgd, illustrates advantages of the proposed algorithm.

3. Experiments are in fully agreement with results of theories, and they have implemented various settings for this: Comparison on number of samples, dependence on training steps, and different parameters.

**Weaknesses:**

1- Improvements over previous work is not very conspicuous. Essentially there is no new point in the paper but having former arguments in a more rigorous manner.

2- Additionally, they don't give any assuring bounds besides two layer neural network, which is the exact model their reference work has considered.

3- There are many non-free hyper parameters in their training process. Having assumed that the number of training steps for the first part of algorithm (T_1) depends on parity size is questionable. This can be the reason why previous work assumed only one step of SGD on sparse data.

4- When attempting to come up with similar bounds besides the covariance loss, which is not common between experimentalists, they only consider hinge loss. And this causes their advantage over previous work to disappear because learning rates gets unbounded.

**Questions:**

1 - Would you please make a clarification on new points of your paper? It seems like your reference(CM23) has covered most of the claims as in your paper.

2- I suggest to include other tasks that curriculum would come to be handful in your work.

**Limitations:**

The main limit of this paper is not to go anywhere else other than the parity task. Many papers have investigated the same object before.

---

> ### Author Rebuttal · Authors · 2023-08-09
>
> Please see the global response for the differences with the previous work [CM23]. We address the rest of the remarks and questions in the list below.
>
> - Q1. Does $T_1$ depend on $k$?
>
> A1. $T_1$ depends on $\mu^k$, through the parameter $L_{\mu}$. This dependence appears in the 1-step arguments as well, and we believe that it cannot be removed. However, we remark that choosing $\mu$ sufficiently close to 1 (e.g. $\mu =1-\frac{1}{d}$ or $\mu =1-\frac{1}{k}$) allows bounding  $\mu^k$ independently from $k$.
>
> - Q2. Hinge Loss learning rate is unbounded.
>
> A2. We believe that Theorem 3 could be extended to include the setting with hinge loss. This however would require a more complicated proof (e.g. by adding a second $l_\infty $ projection to the iterations of SGD).
>
> - Q3. How does the proposed curriculum method generalize to other tasks?
>
> A3. We empirically investigated the suggested curriculum method for Boolean functions other than parity targets (see the last part of Section 5). We have shown that the suggested curriculum method can still be beneficial for learning or weak learning of some Boolean functions. However, we have also shown that there are functions for which the proposed curriculum is not helpful. Based on these experiments, we put forward the conjecture that curriculum can be beneficial in learning of the lowest degree monomials of a function and therefore (based on the structure of the function) it might be helpful or not. Beyond the Boolean functions, we believe similar observations can be found for functions on the Gaussian inputs.  We believe the most analogous real-world application of our setup is the use of input length in NLP/reasoning tasks (note that here $+1$ is the identity element so it can be also viewed as the padding element and the number of $-1$’s can be seen as length – see related literature).

---

> > ### Comment · Reviewer_RLGW · 2023-08-20
> >
> > Thanks for the clarification on the results of this paper. Having reviewed the contributions of this paper and the rebuttal by the authors, I decided to improve my rating and give this a weak accept instead.

---

> > > ### Author Response · Authors · 2023-08-20
> > >
> > > Thank you for considering the rebuttal and increasing your score.

---

### Author Rebuttal · Authors · 2023-08-09

We thank all reviewers for their constructive comments. We address the remarks and questions in the lists below.

## Comparison to [CM23]
Our result is not only an improvement of [CM23] in terms of having more natural training settings and hyperparameters (i.e., we use SGD with bounded batch size and bounded learning rate in Theorem 3), it is also establishing a separation between learning with and without curriculum on a common sampling distribution, while [CM23] does not.

More specifically:
1. We prove a separation between curriculum training and standard training on a common dataset (sampled from $D_{mix}$) in contrast to [CM23], where curriculum training involves both sparse and dense inputs and standard training involves dense inputs only. To the best of our knowledge, our paper is the first work that establishes a rigorous separation between training on adequately ordered samples and randomly ordered samples drawn from the same distribution, and our theorem gives a condition on the mixture under which this separation holds (interestingly the separation does not take place for all mixtures parameters).
2. We prove a positive result for layer-wise curriculum SGD with or without noise with bounded batch size and bounded learning rate (Theorem 3). This involves separating the dynamics into a drift term and a martingale term and adequately bounding each contribution. While such drift-martingales techniques appeared recently in previous works for spherical and Gaussian inputs, to the best of our knowledge, none of the references mentioned in the paper consider Boolean inputs and parity targets. In contrast, [CM23] makes a 1-step argument with an unconventionally large learning rate to show that correlations with the support can be obtained in the hidden layer, without tackling the full dynamics.
3. The martingale proof technique allows us to obtain a tighter upper bound on the number of samples needed to learn with curriculum, compared to the 1-step argument used in [CM23]. In particular, in Theorem 3 we prove that with curriculum we can learn with $\tilde O(d/\rho)$ samples, while applying the same proof technique of [CM23, Theorem 4] allows to obtain an upper bound of only $\tilde O(d^2/\rho)$ samples.
4. We prove a non-trivial negative bound for standard training on inputs sampled from $D_{mix}$, while the negative part of the separation of [CM23] relies on previously established lower bounds for learning parities on uniform inputs. We emphasize that our lower bound holds for any fully connected architecture of any depth, with any activation, any learning rate schedule (e.g., layer-wise) and permutation-invariant initialization.

## Further comments on curriculum and standard training in the theoretical results (Reviewer 8Apd)

- Q7: Difference between currriculum and standard learning.

A7: Batch size: indeed our negative result holds only for noisy-SGD with large batch size. We are not aware of any technique for proving lower bounds for SGD with small batch size. Our positive result holds for the same setting as our negative result AND for other settings as well, such as SGD without noise and any batch size. Thus, our theoretical separation holds for noisy-SGD with large batch size. On the other hand, our experiments use SGD with standard batch size for both curriculum and standard training.

Proportion of easy examples: Our separation holds for small $\rho$, thus for datasets with a small fraction of easy examples. In Theorem 3, we assume the number of easy examples to be at least $\tilde \theta(d)$, which implies that our dataset must be of size at least $\tilde \theta(d/\rho)$. We remark that we do not assume a large proportion of easy examples, we assume instead a large enough dataset (indeed we prove an upper bound on the number of hard samples needed, but the algorithm can use more).

Theoretical results for large $\rho$: Indeed, our experiments surprisingly show that curriculum can be harmful for datasets with many easy samples (at least in terms of number of training steps needed to learn). This case is indeed not covered by our theoretical results. While we do not have a negative result for curriculum training, Theorem 3 gives, in the setting considered, an upper bound on the number of training steps needed to learn with curriculum that does not depend on $\rho$. On the other hand, if we try to apply the footprints of the proof of Theorem 3 to standard training, we would get an upper bound that decreases with $\rho$ (it would depend on the expectation of a k-parity under $D_{mix}$). We do not expect rigorous arguments for  such bounds to be easy extensions of our work.

---

### Decision · Program_Chairs · 2023-09-21

**Decision:**

Accept (poster)

**Comment:**

The paper builds on prior work [CM23] towards understanding curriculum learning using parity functions. Compared to [CM23], they provide tighter convergence analysis under more natural settings, in particular, small batch SGD compared to one-step large batch gradient descent, and bounded learning rates. The paper also has a new negative result showing a separation between curriculum and standard training.

The main concern that came up in the reviews was the novelty in comparison to [CM23] which the authors mostly addressed to the satisfaction of the reviewers. As most reviewers agree, this work provides new theoretical guarantees for the benefits of curriculum learning over standard training which would be of interest to the NeurIPS community. Therefore, I recommend accepting the paper. I encourage the authors to include the feedback from the reviewers as they prepare the camera-ready. In particular, it would help to add the discussion from their rebuttal on the comparison with [CM23], include clarifications to questions that came up for the reviewers, and be more transparent about the limitations of their technical results.